# Reverse oxygen spillover triggered by CO adsorption on Sn-doped Pt/TiO$_2$ for low-temperature CO oxidation

Jianjun Chen [1,4], Shangchao Xiong [2,4] ✉, Haiyan Liu[1], Jianqiang Shi[1], Jinxing Mi[1], Hao Liu[1], Zhengjun Gong [2], Laetitia Oliviero [3], Françoise Maugé [3] & Junhua Li [1] ✉

The spillover of oxygen species is fundamentally important in redox reactions, but the spillover mechanism has been less understood compared to that of hydrogen spillover. Herein Sn is doped into TiO$_2$ to activate low-temperature (<100 °C) reverse oxygen spillover in Pt/TiO$_2$ catalyst, leading to CO oxidation activity much higher than that of most oxide-supported Pt catalysts. A combination of near-ambient-pressure X-ray photoelectron spectroscopy, in situ Raman/Infrared spectroscopies, and ab initio molecular dynamics simulations reveal that the reverse oxygen spillover is triggered by CO adsorption at Pt$^{2+}$ sites, followed by bond cleavage of Ti-O-Sn moieties nearby and the appearance of Pt$^{4+}$ species. The O in the catalytically indispensable Pt-O species is energetically more favourable to be originated from Ti-O-Sn. This work clearly depicts the interfacial chemistry of reverse oxygen spillover that is triggered by CO adsorption, and the understanding is helpful for the design of platinum/titania catalysts suitable for reactions of various reactants.

Platinum group metals supported on oxides are widely used in environmental catalysis, electrocatalysis, and energy-related hydrogenation processes[1–4]. To understand the mechanisms, interfacial chemistry of catalytic reactions over noble metals is of great importance in the development of catalytic science. The concept of strong metal-support interaction (SMSI) effect has been widely used to describe and/or interpret phenomena of electronic interaction, as well as the stabilization/destabilization of metals on support materials[5,6]. For instance, Ding et al.[7] synthesized single-atom Pt/CeO$_2$ catalysts via oxidative and non-oxidative dispersions, and found significant differences in the CO catalytic oxidation activity of single-atom Pt in different coordination environments. Another important concept is the transport of adsorbates and/or intermediates through metal support interfaces, known as spillover of active species, such as hydrogen and oxygen entities. For hydrogen spillover, the fundamentals have been well documented[8] and widely applied in catalyst design[9,10]. By contrast, much less

attention has been given to oxygen spillover, and the related information is limited.

Recently reverse oxygen spillover (ROS) has opened new opportunities in improving the activity, selectivity, and stability of the catalyst systems with Ce-based supports due to the excellent oxygen mobility of ceria. For example, Hensen and co-workers[11] investigated the interfacial dynamics of Pd-CeO$_2$ catalyst in CO oxidation, and found that the surface oxidized Pd species showed high resistance against sintering ascribable to the extent ROS, whereas the ionic Pd species in a Pd-CeO$_2$ system without ROS underwent swift reduction and agglomeration. Also, by varying the particle size of CeZrO$_4$ support of the Co/Ce-Zr catalysts, Hensen et al.[12] observed facile formation of oxygen vacancies ascribable to ROS, and upon CO/CO$_2$ exposure the filling of vacancies by oxygen atoms via oxygen spillover. The migration of oxygen species enables the stabilization of cobalt metal nanoparticles and CO$_2$ activation, thus playing essential roles in various CO$_2$

[1]State Key Joint Laboratory of Environment Simulation and Pollution Control, School of Environment, Tsinghua University, Beijing 100084, PR China. [2]Faculty of Geosciences and Environmental Engineering, Southwest Jiaotong University, Chengdu 610031, PR China. [3]Laboratoire Catalyse et Spectrochimie, ENSI-CAEN, Université de Caen, CNRS, 6 bd du Maréchal Juin, 14050 Caen, France. [4]These authors contributed equally: Jianjun Chen, Shangchao Xiong. ✉e-mail: xiongshangchao@swjtu.edu.cn; lijunhua@tsinghua.edu.cn

hydrogenation reactions. Theoretical calculations were also employed to investigate the ROS in $CeO_2$-based catalysts. Combining DFT predictions and resonant photoelectron spectroscopy, Vayssilov et al.[13] proposed that the ROS process occurred on nanostructured ceria rather than on bulk ceria. In their study, a model Pt-$CeO_2$-film catalyst was designed and the generation of $Ce^{3+}$ was used to indirectly indicate the transfer of O from support to the active Pt sites. It should be noted that the spillover of oxygen species is fast and its rate can compete with the rate of reaction. Therefore, the direct observation of such a phenomenon is difficult and challenging.

Because the oxygen mobility of $TiO_2$ is much lower than that of $CeO_2$, systemic ROS study on Pt/Titania catalyst is extremely rare despite the related catalytic behavior is known. By a potential dynamic sweep method[14], Lin studied oxygen spillover and back spillover over Pt/$TiO_2$ working electrode. It was found that the transport of oxygen species is brought about not only on the surface but also significantly in the $TiO_2$ crystal lattice. Nevertheless, only current-potential profiles of Pt/$TiO_2$ electrode were reported and the related interface chemistry was not illustrated in detail. More importantly, the effect of electricity introduction into the system cannot be ignored, making the information obtained by electrochemical methods, to a certain extent, irrelevant to thermocatalysis. Therefore, it is appropriate despite challenging to activate the ROS process in $TiO_2$-based supports and to understand the ROS dynamics of Pt/Titania interfaces in heterogeneous catalysis.

Since $SnO_2$ possess a similar structure as rutile $TiO_2$[15], doping Sn will not change significantly the bulk construction but create asymmetric oxygens ($M_1$-O-$M_2$) in $TiO_2$. Such altering of oxygen symmetry probably increases its mobility and thus benefits the ROS process. Therefore, in the present work, we modulated the rutile $TiO_2$ by Sn doping to activate the oxygen in $TiO_2$ support, and illustrated the rich interfacial chemistry of reverse oxygen spillover from Sn-doped $TiO_2$ ($SnTiO_2$) to Pt sites in low-temperature (100 °C) CO oxidation. Comparing with the reference catalysts, namely, anatase-supported and rutile-support Pt (denoted herein as Pt/$TiO_2$-A and Pt/$TiO_2$-R, respectively), the fabricated Pt/$SnTiO_2$ was intrinsically 6–12 times more active and there was no sight of deactivation in a test of seven days, based on the USDRIVE's protocol[16] in the presence of 10% $H_2O$ and 100 ppm $SO_2$. The revealed interface chemistry suggested that the reverse oxygen spillover was triggered by CO adsorption on Pt sites followed by the appearance of $Pt^{4+}$ species. With the cleavage of Ti-O-Sn bonds in the vicinity of CO-adsorbed Pt sites, there is the availability of O species and the formation of catalytically indispensable Pt-O sites. The revealed reactant-adsorption-triggered characteristics of interfacial reverse oxygen spillover can help understand the mechanistic aspects of catalytic reactions that are different in reactants, as well as those of photo-electrocatalytic nature such as oxygen evolution and reduction reactions.

## Results

### CO oxidation performance

The absence of heat and mass transfer limitations was verified by Mears criterion[11] (Supplementary Note 1 and Supplementary Fig. S1). The $H_2$ pretreatment temperature and the valence of Pt were discussed by Supplementary Note 2 and Supplementary Fig. S2–3. As displayed in Supplementary Fig. S4, the Pt/$Sn_xTi_{1-x}O_2$ catalysts exhibited satisfactory CO oxidation activities with $T_{90} < 120$ °C. Specifically, Pt/$Sn_{0.2}Ti_{0.8}O_2$ showed the highest activity with CO conversion of -100% at 120 °C, meeting the guideline of "90% conversion of all criteria pollutants at 150 °C" proposed by the U.S. Department of Energy[17]. Fig. 1a shows that the CO oxidation activity of Pt/$Sn_{0.2}Ti_{0.8}O_2$ obviously better than those of Pt/$TiO_2$-R and Pt/$TiO_2$-A with the same Pt loading

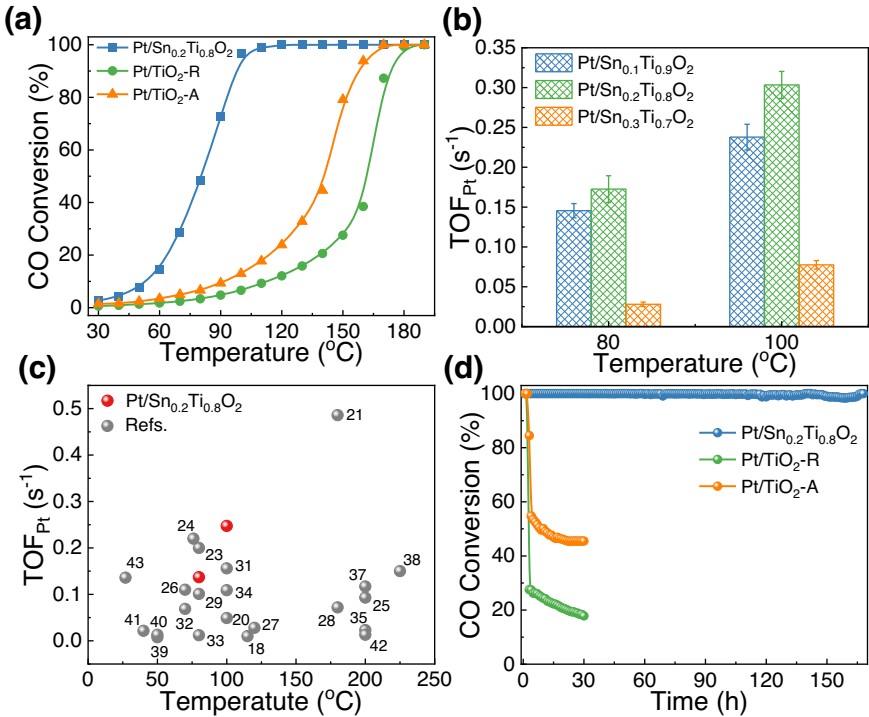

**Fig. 1 | CO oxidation performance over Pt/$Sn_{0.2}Ti_{0.8}O_2$, Pt/$TiO_2$-R, and Pt/$TiO_2$-A pretreated with 5% $H_2$ at 300 °C for 1 hour. a** CO conversion plots under steady-state feed of "1% CO, 1% $O_2$, $N_2$ balance" and GHSV of 60,000 ml $g_{cat}^{-1}$ $h^{-1}$. **b** Turnover frequencies of CO oxidation over Pt sites ($TOF_{Pt}$) measured at 80 °C and 100 °C with CO conversion below 20%. The error bars represent standard deviations. **c** Comparison of $TOF_{Pt}$ between Pt/$Sn_{0.2}Ti_{0.8}O_2$ and the Pt-based catalysts in reported references. **d** Long-term test of sulfur resistance in CO oxidation. Reaction conditions: reaction feed of 1% CO, 1% $O_2$, 10% $H_2O$, 100 ppm $SO_2$, $N_2$ balance; GHSV of 60,000 ml $g_{cat}^{-1}$ $h^{-1}$; and reaction temperature of 240 °C.

(0.5 wt%), indicating that Sn doping plays a critical role in promoting low-temperature CO oxidation over the Pt/$Sn_{0.2}Ti_{0.8}O_2$ catalyst. A comparison between Fig. 1a and Supplementary Fig. S5a revealed that a moderate $H_2$ reduction (5% $H_2$, 300 °C) of the catalysts resulted in obvious improvement of CO oxidation activities over the Pt/$Sn_{0.2}Ti_{0.8}O_2$ catalyst, whereas over Pt/$TiO_2$-R and Pt/$TiO_2$-A the improvement was less obvious. A possible reason is that the reaction pathways for CO oxidation over Pt/$Sn_{0.2}Ti_{0.8}O_2$ and Pt/$TiO_2$ catalysts are different (vide infra).

In agreement with the CO oxidation activities, the activation energies ($E_a$) over all the Pt/$Sn_xTi_{1-x}O_2$ catalysts (especially for Pt/$Sn_{0.2}Ti_{0.8}O_2$) were lower than those over Pt/$TiO_2$-R and Pt/$TiO_2$-A (Supplementary Fig. S6). Moreover, the $E_a$ of the catalysts without $H_2$ pretreatment (Supplementary Fig. S5b) were obviously higher than those of the catalysts with $H_2$ pretreatment (Supplementary Fig. S6), suggesting that the $H_2$ reduction treatment could optimize the active sites and accelerate the CO oxidation process. Even compared with reported Pt-based catalysts[18–43], the Pt/$Sn_{0.2}Ti_{0.8}O_2$ catalyst is much lower in $E_a$ value, except in cases such as 1%wt Pt/CNT-600 and 0.70% wt $Pt_{NPS}$/$TiO_2$−x (Supplementary Fig. S7 and Supplementary Table S1). Through calculation, the turnover frequency of CO conversion over Pt sites ($TOF_{Pt}$) of Pt/$Sn_{0.2}Ti_{0.8}O_2$ is 0.30 $s^{-1}$ at 100 °C (Fig. 1b), exhibiting a level comparable to those of the superior Pt-based catalysts[18–43] (Fig. 1c and Supplementary Table S1). Meanwhile, the reaction order of CO and $O_2$ during CO catalytic oxidation over Pt/$Sn_{0.2}Ti_{0.8}O_2$, Pt/$TiO_2$-R, and Pt/$TiO_2$-A were all slightly higher than 0 (Supplementary Fig. S8), indicating that CO oxidation over these three Pt-based catalysts follows the Mars−van Krevelen (MvK) mechanism, which is typical for reducible oxide-based catalysts[11]. Additionally, the partial orders of CO (or $O_2$) were approximately 0, highlighting the weakened kinetic relevance of CO (or $O_2$) adsorption/activation over Pt/$Sn_{0.2}Ti_{0.8}O_2$, Pt/$TiO_2$-R, and Pt/$TiO_2$-A[22].

On top of the superior CO oxidation performance, the Pt/$Sn_{0.2}Ti_{0.8}O_2$ catalyst shows a satisfactory sulfur resistance ability, giving 100% CO conversion in a span of 7 days even in the presence of 10% $H_2O$ and 100 ppm $SO_2$ (Fig. 1d), demonstrating that Pt/$Sn_{0.2}Ti_{0.8}O_2$ could operate reasonably well under complicated conditions harsher than those of USDRIVE's protocol[16]. One plausible explanation for the enhanced sulfur resistance is that the introduction of Sn doping results in a notable modification of the coordination environment of Ti within the support structure, thereby influencing the interaction between the active Pt site and the support, leading to improved sulfur resistance[44]. Moreover, harsh pretreatment process (e.g., hydrothermal aging by 10% $H_2O$ at 750 °C for 9 h) could not impair the performance of Pt/$Sn_{0.2}Ti_{0.8}O_2$, which still exhibited CO conversion of ~100% at 200 °C (Supplementary Fig. S9). To sum up, by doping a proper content of Sn into the titania of $TiO_2$-supported Pt catalyst, excellent CO oxidation activity, as well as sulfur resistance ability under complicated conditions, could be acquired.

## Catalyst structure

A series of analyses were conducted to characterize the microstructure of Pt/$Sn_xTi_{1-x}O_2$, Pt/$TiO_2$-R, and Pt/$TiO_2$-A. The X-ray diffraction (XRD) patterns (Supplementary Fig. S10) suggest that the reflection peaks of Pt/$Sn_xTi_{1-x}O_2$ and Pt/$TiO_2$-R catalysts could be assigned to rutile $TiO_2$ (JCPDS: #21−1276), whereas those of Pt/$TiO_2$-A to anatase $TiO_2$ (JCPDS: #21−1272). No reflections corresponded to Pt species could be observed plausibly due to the low content and well dispersion of Pt species on the support. Further Rietveld refinements of the XRD patterns (Supplementary Fig. S11 and Supplementary Table S2) suggest that the doped Sn was embedded in the $TiO_2$ crystal structure with the substitution of Ti in an appropriate theoretical dosage. Interestingly, despite different in phase structure, the Pt/$Sn_xTi_{1-x}O_2$ and Pt/$TiO_2$-A catalysts are similar in texture parameters (Supplementary Fig. S12 and Supplementary Table S3).

The microchemical state was studied by X-ray absorption near edge spectroscopy (XANES). As shown in Fig. 2a, the peaks at 530.6 eV and 533.3 eV of O K-edge XANES spectra could be assigned to the $T_{2g}$ and $E_g$ states of $TiO_2$[45,46]. The O K-edge XANES spectra of Pt/$TiO_2$-R and Pt/$TiO_2$-A correspond well to that of $TiO_2$, whereas the O K-edge XANES spectrum of Pt/$Sn_{0.2}Ti_{0.8}O_2$ is more distorted[1], suggesting the doped Sn could significantly change the electronic interactions between oxygen and metal. Fig. 2b gives the XANES spectra of Pt $L_3$-edge over Pt/$Sn_{0.2}Ti_{0.8}O_2$, Pt/$TiO_2$-R, and Pt/$TiO_2$-A, and the reference spectra of Pt and $PtO_2$ are shown in Supplementary Fig. S13a. The Pt $L_3$-edge XANES spectra of Pt/$Sn_{0.2}Ti_{0.8}O_2$, Pt/$TiO_2$-R, and Pt/$TiO_2$-A are almost identical. However, the edge position and edge jump of Pt/$Sn_{0.2}Ti_{0.8}O_2$ are both slightly lower than those of Pt/$TiO_2$-R and Pt/$TiO_2$-A, indicating a slightly more reduced Pt state for Pt/$Sn_{0.2}Ti_{0.8}O_2$[47,48]. A similar conclusion can be drawn from the data of $k^2$ weighted Fourier transform-extended X-ray absorption fine structure (FT-EXAFS) shown in Fig. 2c and Supplementary Fig. S13b, Supplementary Fig. S14 and Supplementary Table S4. All the samples contained a strong peak at ~1.8 Å ascribable to Pt−O bond in $PtO_x$[48,49], among which that of Pt/$Sn_{0.2}Ti_{0.8}O_2$ is the shortest. In addition, the intensities of the Pt−O peaks of Pt/$Sn_{0.2}Ti_{0.8}O_2$, Pt/$TiO_2$-R, and Pt/$TiO_2$-A (Supplementary Fig. S14) are all much weaker than that of reference $PtO_2$ (Supplementary Fig. S13b); with due consideration to the weaker absorption edge intensity and shorter Pt-O bond length, it is deduced that the Pt valence of all the samples is lower than that of $PtO_2$. It is noted that there is weak Pt−O−Sn bonding at ~3.0 Å of Pt/$Sn_{0.2}Ti_{0.8}O_2$ (Fig. 2c), suggesting interaction feasibility of Pt with the $Sn_xTi_{1-x}O_2$ support. The Pt−O environments of the Pt/$Sn_{0.2}Ti_{0.8}O_2$ were more disorder in comparison to that of Pt/$TiO_2$-R and Pt/$TiO_2$-A. Moreover, although Pt-Pt bond was taken into consideration when fitting the data, the low Pt-Pt coordination number (CN < 2, see Supplementary Table S4) for all the samples suggested the extremely low metallic nature of Pt presumably due to the moderate reduction conditions (300 °C and 5% $H_2$), which are consistent with the results of XPS characterizations (vide infra).

The scanning electron microscopic (SEM) images of Pt/$Sn_{0.2}Ti_{0.8}O_2$, Pt/$TiO_2$-R, and Pt/$TiO_2$-A (Supplementary Fig. S15) display micro morphologies of particle accumulation. The average particle size of Pt/$Sn_{0.2}Ti_{0.8}O_2$ is similar to that of Pt/$TiO_2$-A, but smaller than that of Pt/$TiO_2$-R. High-resolution transmission electron microscopic (HR-TEM) images (Supplementary Fig. S16) reveal that the exposed crystal planes of Pt/$Sn_{0.2}Ti_{0.8}O_2$ and Pt/$TiO_2$-R are mainly the (110) planes of rutile $TiO_2$, whereas the major exposed planes of Pt/$TiO_2$-A are the (101) planes of anatase $TiO_2$. Energy dispersive X-ray spectrometry (EDX) mapping (Supplementary Fig. S17) was conducted to investigate the element distribution of Pt/$Sn_{0.2}Ti_{0.8}O_2$ and Pt/$TiO_2$-R. Obviously, there is even dispersion of Sn, Ti, and O elements, but Pt was hardly observed in the mapping due to its low amount (0.5 wt%). To directly observe the microstate of loaded Pt species, we employed spherical aberration-corrected scanning transmission electron microscopy (SAC-STEM). The high-angle annular dark-field (HAADF) and the corresponding bright field (HAABF) images are shown in Fig. 2d and Supplementary Fig. S18. It should be noted that the Pt species in Pt/$Sn_{0.2}Ti_{0.8}O_2$ are hardly observed directly by the SAC-STEM images because of the closeness in atomic numbers between Sn (50) and Pt (78). However, we could still observe the microstate of Pt species by comparing the HAADF and HAABF images, which are circled in Fig. 2d and Supplementary Fig. S18. The Pt species in all the samples were in the form of nano clusters, and the average diameter of the Pt clusters in Pt/$Sn_{0.2}Ti_{0.8}O_2$, Pt/$TiO_2$-R, and Pt/$TiO_2$-A was 1.2, 1.8, and 0.9 nm, respectively.

It is generally accepted that the CO oxidation process over various reducible supported Pt catalysts follows a MvK reaction mechanism[50]. The CO adsorbed on Pt nanoclusters reacts with active lattice oxygen species ($O_{latt}$) of the supports, and there is no involvement of competitive adsorption between CO and $O_2$[19,25]. Therefore, a transient CO

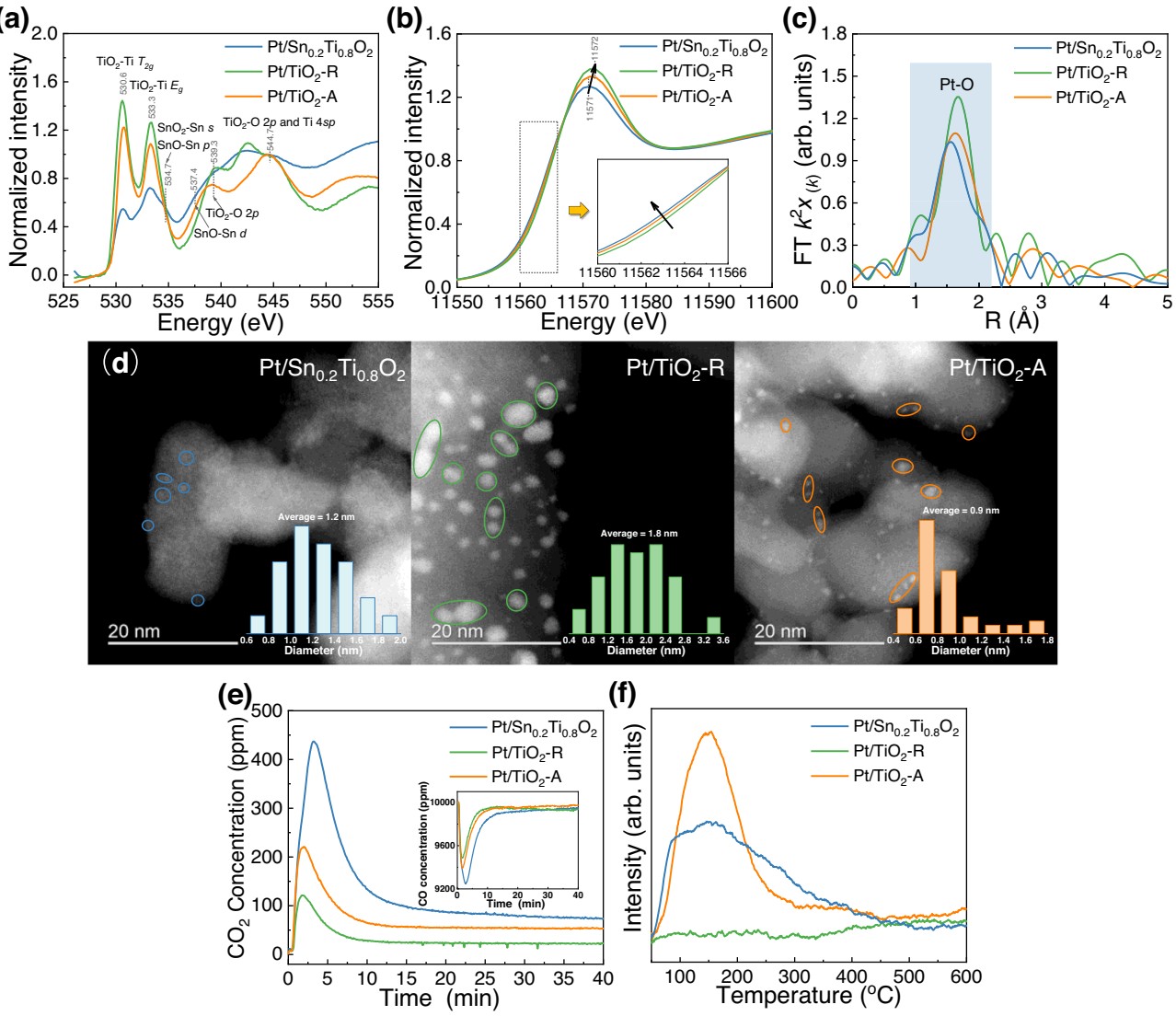

**Fig. 2 | Static characterization of Pt/Sn$_{0.2}$Ti$_{0.8}$O$_2$, Pt/TiO$_2$-R, and Pt/TiO$_2$-A pretreated with 5% H$_2$ at 300 °C. a** XANES spectra of O K-edge. **b** XANES spectra of Pt L$_3$-edge. **c** Magnitude component of the k$^2$ weighted FT-EXAFS data of Pt L$_3$-edge. **d** SAC-STEM HAADF images; inset shows the particle size distribution of Pt clusters. **e** CO$_2$ generation and corresponding CO concentration (inset) as a function of time during the transient CO oxidation without O$_2$ supply (1% CO/N$_2$) at 100 °C. **f** O$_2$-TPD profiles.

oxidation study without continuous O$_2$ supply was conducted at 100 °C; thus the amount of CO$_2$ generation could represent the available amount of active O. Meanwhile, the variation of CO concentration during the study was recorded. As shown in Fig. 2e, the amounts of CO$_2$ generation follow a decreasing order of Pt/Sn$_{0.2}$Ti$_{0.8}$O$_2$ > Pt/TiO$_2$-A > Pt/TiO$_2$-R, suggesting that Pt/Sn$_{0.2}$Ti$_{0.8}$O$_2$ has the highest amount of active O$_{latt}$. A similar order was observed when the study was conducted at 200 °C (see Supplementary Fig. S19), suggesting that Pt/Sn$_{0.2}$Ti$_{0.8}$O$_2$ is the highest in terms of the availability of active O disregard of the variation of reaction temperature. Interestingly, the results of O$_2$ temperature-programmed desorption (O$_2$-TPD) over the three catalysts reveal that the extents of O$_2$ desorption follow a decreasing order of Pt/TiO$_2$-A > Pt/Sn$_{0.2}$Ti$_{0.8}$O$_2$ > Pt/TiO$_2$-R (Fig. 2f), implying that Pt/TiO$_2$-A has the highest amount of active oxygen. The implication differs from that of transient CO oxidation study (Fig. 2e). To explain the inconsistency, it is assumed that there was in situ generation of active O species when Pt/Sn$_{0.2}$Ti$_{0.8}$O$_2$ was exposed to CO, and these in situ generated O species upon CO introduction cannot be measured by O$_2$-TPD study. Therefore, a series of in situ studies were performed to verify this assumption.

## Existence of reverse O spillover

In situ near-ambient pressure X-ray photoelectron spectroscopy (NAP-XPS) is an effective technique to investigate the chemical behavior of catalyst surfaces. The in situ NAP-XPS Pt 4$f$ spectra were recorded at 100 °C of the reduced samples and after it were exposed to O$_2$, CO + O$_2$, and CO in turn. It should be noted that the Ti 3 s satellite (at ~75 eV) overlaps with the Pt 4$f$ peaks, causing complication in the deconvolution of the Pt 4$f$ profiles in cases such as Pt/Sn$_{0.2}$Ti$_{0.8}$O$_2$, Pt/TiO$_2$-R, and Pt/TiO$_2$-A catalysts[48,51]. Therefore, the peak area ratio of Pt 4$f_{5/2}$ to 4$f_{7/2}$ was strictly fixed as 3: 4, and the full width of half maximum (FWHM) of all the Pt$^{2+}$ and Pt$^{4+}$ peaks was set to be identical during the deconvolution process. In Fig. 3, the peaks at ~72.2 eV and ~74.8 eV could be assigned to 4$f_{7/2}$ and 4$f_{5/2}$ signals of Pt$^{2+}$ species, whereas the peaks at ~74.4 eV and ~77.8 eV to 4$f_{7/2}$ and 4$f_{5/2}$ signals of Pt$^{4+}$ species, respectively[48]. There was no detection of peaks attributable to metallic Pt species in the entire in situ NAP-XPS study. After moderate H$_2$ pretreatment, only Pt$^{2+}$ species could be observed (Fig. 3a), indicating that the Pt species on the surface of Pt/Sn$_{0.2}$Ti$_{0.8}$O$_2$ was mainly PtO. Moreover, no Pt$^{4+}$ species could be observed after further O$_2$ treatment, suggesting that the PtO was stable upon the O$_2$

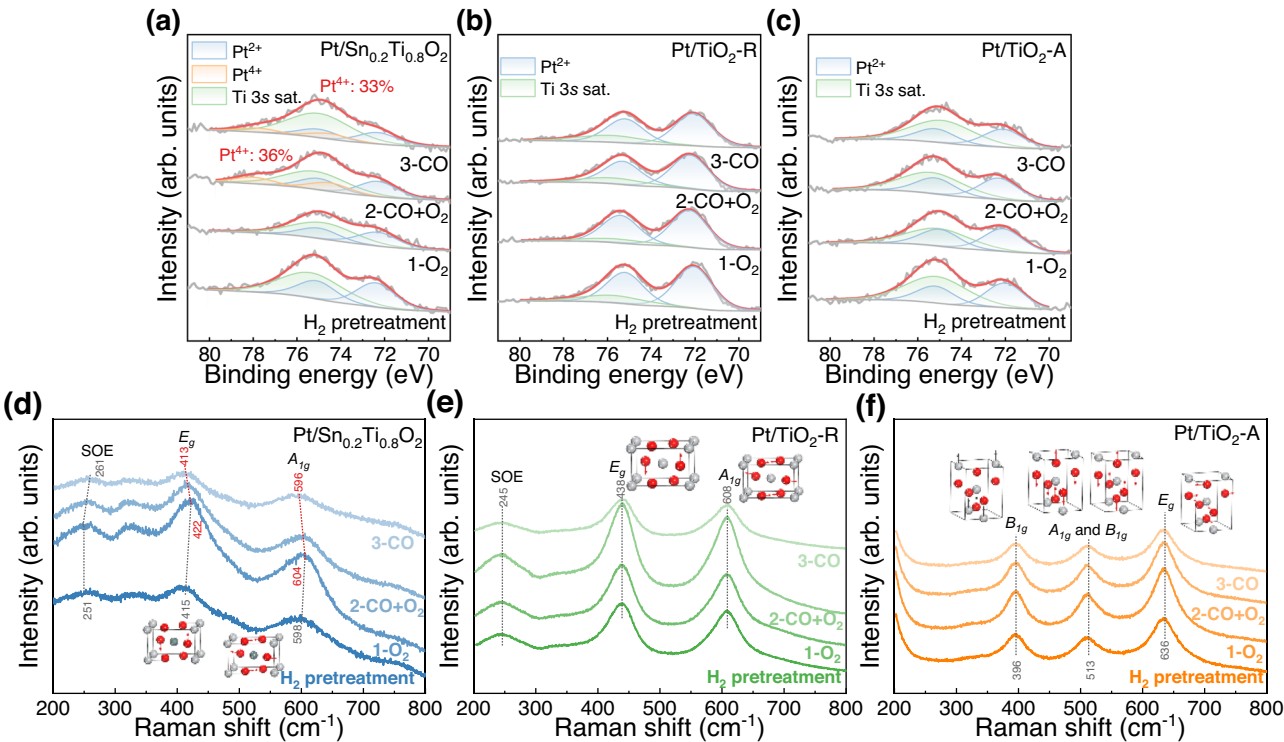

**Fig. 3 | In situ NAP-XPS and in situ Raman studies of Pt/Sn$_{0.2}$Ti$_{0.8}$O$_2$, Pt/TiO$_2$-R and Pt/TiO$_2$-A pretreated with 5% H$_2$ at 300 °C. a–c** In situ NAP-XPS Pt 4$f$ spectra recorded at 100 °C after (**a**) Pt/Sn$_{0.2}$Ti$_{0.8}$O$_2$, (**b**) Pt/TiO$_2$-R, and (**c**) Pt/TiO$_2$-A was exposed to 1 mbar of O$_2$, 0.5 mbar of CO with 0.5 mbar of O$_2$, and 1 mbar of CO in turn. **d–f** In situ Raman spectra recorded at 100 °C after (**d**) Pt/Sn$_{0.2}$Ti$_{0.8}$O$_2$, (**e**) Pt/TiO$_2$-R, and (**f**) Pt/TiO$_2$-A was exposed to 1% O$_2$, 1% CO + 1% O$_2$, and 1% CO in turn under ambient pressure. The term SOE means the second-order effect.

exposure. However, there was the generation of Pt$^{4+}$ species upon the introduction of CO + O$_2$ over Pt/Sn$_{0.2}$Ti$_{0.8}$O$_2$, and the Pt$^{4+}$ species existed only in the presence of CO. It should be emphasized that CO is a reducing gas, and could be oxidized to CO$_2$ by the O$_{latt}$ of Pt/Sn$_{0.2}$Ti$_{0.8}$O$_2$ without the introduction of gaseous O$_2$ (see Fig. 2e). In Fig. 3a, the oxidation of Pt$^{2+}$ to Pt$^{4+}$ occurred concurrently with CO oxidation, implying that there was first the transfer of O$_{latt}$ species to Pt sites, and then the oxidation of CO to CO$_2$ by the O$_{latt}$ species at the Pt sites. However, when O$_{latt}$ species transferred to Pt sites, the decrease of valence state of Ti and Sn species was hardly observed by the in situ NAP-XPS technique due to the content difference of about 2 orders of magnitude between Pt and carrier elements (Ti, Sn, and O) (see Supplementary Note 3). As for Pt/TiO$_2$-R and Pt/TiO$_2$-A, only Pt$^{2+}$ species could be observed upon all the exposures (Fig. 3b, c), suggesting the absence of reverse O spillover (ROS) over Pt/TiO$_2$-R and Pt/TiO$_2$-A. The overall results evidence that the doped Sn promoted the migration of O$_{latt}$ in the TiO$_2$ support, leading to the occurrence of the ROS process upon CO introduction.

Raman spectra were also in situ recorded at 100 °C after the reduced catalysts were exposed to O$_2$, CO + O$_2$, and CO in turn. As shown in Fig. 3d, e, the Raman spectra of Pt/Sn$_{0.2}$Ti$_{0.8}$O$_2$ and Pt/TiO$_2$-R both exhibit three peaks at ~604, ~422, and ~251 cm$^{-1}$, ascribable to the $A_{1g}$ and $E_g$ vibration modes and the second-order effect (SOE) of rutile TiO$_2$, respectively[52]. Interestingly, the $A_{1g}$ and $E_g$ vibration modes of Pt/Sn$_{0.2}$Ti$_{0.8}$O$_2$ shift to lower wavenumbers after exposure to CO + O$_2$, and more obviously with the sole introduction of CO. The $A_{1g}$ vibration mode is derived from the symmetric stretching of O-Ti(Sn)-O in the (110), (1̄10) and (001) plane, and the $E_g$ vibration mode represents asymmetric bending of O-Ti(Sn)-O in the (110) plane. The shifts to lower wavenumbers of $A_{1g}$ and $E_g$ vibration modes indicate weakened bond strength of O-Ti(Sn)-O[52], implying higher mobility of O$_{latt}$ in TiO$_2$, thereby contributing to the ROS during CO oxidation on the surface of Pt/Sn$_{0.2}$Ti$_{0.8}$O$_2$ catalyst. In the cases of Pt/TiO$_2$-R, there was no obvious

change of peak positions with the introduction of CO, indicating that there is no Raman-visible change of O-Ti-O bond strength. As for Pt/TiO$_2$-A (Fig. 3f), the peaks at 636, 513, and 396 cm$^{-1}$ were assigned to the vibration modes of $E_g$, $A_{1g}$/$B_{1g}$ doublet, and $B_{1g}$, respectively[53]. The signal positions of these vibration modes also remain unchanged when Pt/TiO$_2$-A was exposed to CO, indicating that the O-Ti-O bond strength was also kept unchanged during the CO oxidation reaction at 100 °C.

Spectra of diffuse reflectance infrared Fourier transform spectroscopy (DRIFTS) were in situ recorded at 30–100 °C after having the Pt-based catalysts exposed to CO + O$_2$ (Fig. 4). There was the appearance of three peaks at 2172, 2117, and ~2065 cm$^{-1}$ after the exposure. The peaks at 2172 cm$^{-1}$ and 2117 cm$^{-1}$ were assigned to gaseous and weakly adsorbed CO, and the peaks at ~2065 cm$^{-1}$ to adsorbed CO on semi-oxidized Pt species (Pt$^{\delta+}$-CO)[11,43,54,55], which are sensitive to factors such as Pt dispersion, Pt microstructure (i.e., steps, crystal planes and so on) and charge transfer nearby the Pt sites[48]. At higher temperature, more Pt sites would participate in the reaction cycle of CO catalytic oxidation. Thus, the average valence state of Pt should decrease with increasing reaction temperature, leading to deviation of Pt$^{\delta+}$-CO infrared peak. As shown in Fig. 4a–c, when the reaction temperature was increased from 30 to 100 °C, the peaks of Pt$^{\delta+}$-CO over Pt/TiO$_2$-R and Pt/TiO$_2$-A shifted to lower wavenumbers, while an opposite trend was observed over Pt/Sn$_{0.2}$Ti$_{0.8}$O$_2$, which could be caused by ROS. To verify this conception, in situ NAP-XPS spectra were acquired under the same reaction conditions of the DRIFTS experiments. As shown in Fig. 4d–f, only Pt/Sn$_{0.2}$Ti$_{0.8}$O$_2$ possessed the Pt$^{4+}$ species after the introduction of CO + O$_2$, and the Pt$^{4+}$ content increased with the increase of reaction temperature, suggesting that ROS was more significant with the increase of CO oxidation rate over Pt/Sn$_{0.2}$Ti$_{0.8}$O$_2$.

**The ROS investigated by AIMD**

The configurations of Pt/Sn$_{0.2}$Ti$_{0.8}$O$_2$ and Pt/TiO$_2$-R employed in Density Functional Theory (DFT) simulation were obtained from an ab

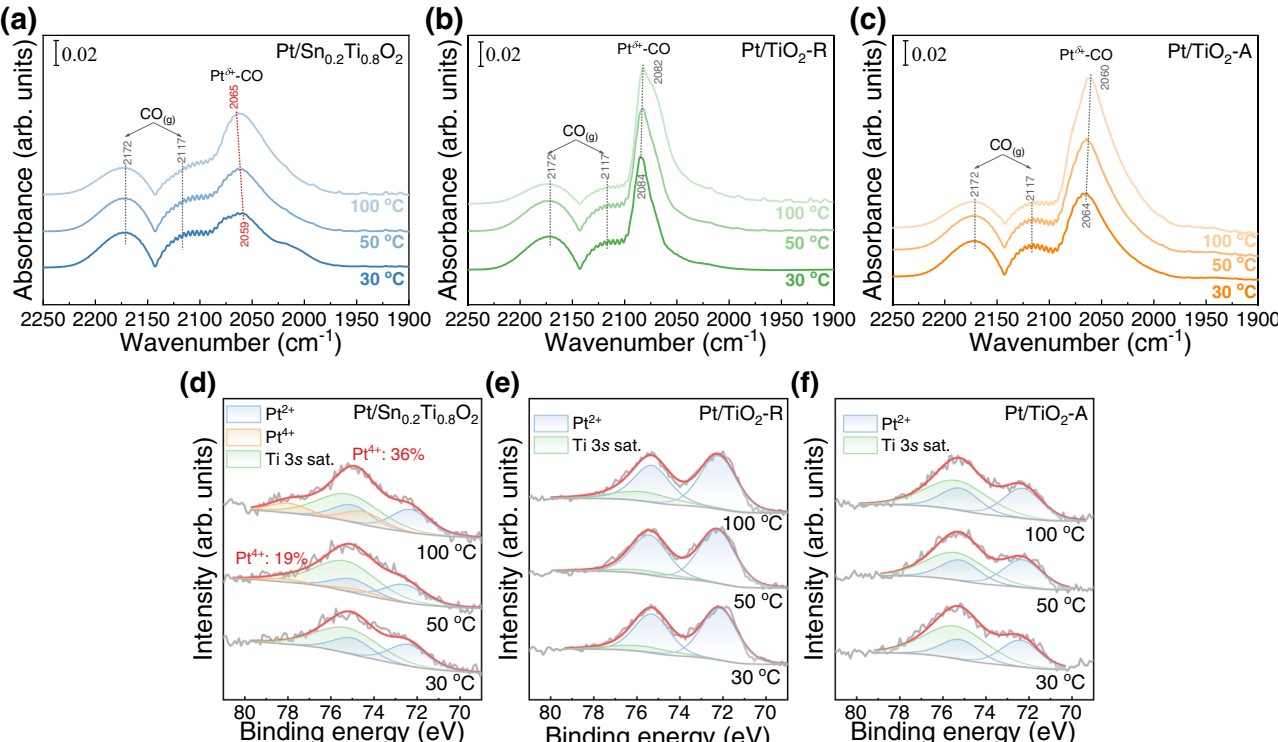

**Fig. 4 | In situ DRIFTS and in situ NAP-XPS spectra of Pt/Sn$_{0.2}$Ti$_{0.8}$O$_2$, Pt/TiO$_2$-R, and Pt/TiO$_2$-A recorded at 30–100 °C after exposure to CO + O$_2$. a–c** In situ DRIFTS spectra recorded at 30–100 °C after (**a**) Pt/Sn$_{0.2}$Ti$_{0.8}$O$_2$, (**b**) Pt/TiO$_2$-R, and (**c**) Pt/TiO$_2$-A was exposed to 1% CO + 1% O$_2$ at a designated temperature under ambient pressure. **d–f** In situ NAP-XPS Pt 4$f$ spectra recorded at 30–100 °C after (**d**) Pt/Sn$_{0.2}$Ti$_{0.8}$O$_2$, (**e**) Pt/TiO$_2$-R, and (**f**) Pt/TiO$_2$-A was exposed to 0.5 mbar of CO with 0.5 mbar of O$_2$ at a designated temperature.

initio molecular dynamics (AIMD) simulation combined with geometrical optimization (see Supplementary Note 4, Supplementary Figs. S20–21, and Supplementary Table S5). Then the charge density difference study was performed on Pt/Sn$_{0.2}$Ti$_{0.8}$O$_2$ and Pt/TiO$_2$-R with the loading of Pt$_4$O$_4$ clusters (Supplementary Fig. S22) and the adsorption of CO (Fig. 5a, b). The loading of Pt$_4$O$_4$ clusters led to an increase of charge density of O in Sn-O-Ti around Pt, which was further enhanced after the subsequent adsorption of CO over Pt/Sn$_{0.2}$Ti$_{0.8}$O$_2$ (Fig. 5a and Supplementary Fig. S22a). The increase of charge density of O suggests an increase of O mobility, benefiting the ROS in Pt/Sn$_{0.2}$Ti$_{0.8}$O$_2$ during CO oxidation reaction. As for Pt/TiO$_2$-R, there is no obvious change of "O" charge density in the Ti-O-Ti around Pt upon the loading of Pt$_4$O$_4$ clusters as well as upon the adsorption of CO (Fig. 5b and Supplementary Fig. S22b).

An AIMD simulation was conducted to further illustrate the ROS process. A CO molecule was adsorbed on the Pt sites of Pt/Sn$_{0.2}$Ti$_{0.8}$O$_2$ and Pt/TiO$_2$-R. Because of the relatively short time scale of AIMD (20 ps in this study), the sampling of AIMD is only suitable to fast events of low-energy barrier, and the observation of slow processes is excluded. Therefore, raising the simulation temperature can result in quick exploration of a large number of phase spaces, and thus the AIMD simulation temperature was performed at 700 K (427 °C), which is slightly lower than the synthesis temperature of SnTiO$_2$ support[56]. Figure 5c, d show the variation of atomic distance between Ti and O (Ti-O), Sn and O (Sn-O), and Pt and O (Pt-O) as a function of simulation time over Pt/Sn$_{0.2}$Ti$_{0.8}$O$_2$ and Pt/TiO$_2$-R. The atomic distance of Ti-O, Sn-O and Pt-O stays at 2-3 Å during the first 6 ps of AIMD simulation (Fig. 5c), indicating that the O in Sn-O-Ti kept bonding with the Pt of Pt$_4$O$_4$ clusters. Then, the atomic distance of Ti-O and Sn-O quickly increased at 6–9 ps, while the atomic distance of Pt-O remained unchanged. The result indicates cleavage of Ti-O and Sn-O bonds in Sn-O-Ti, subsequently leading to the reverse flow of O towards the Pt sites. Finally, such structure remained stable to the end of the AIMD

simulation. The whole AIMD simulation process of Pt/Sn$_{0.2}$Ti$_{0.8}$O$_2$ can be directly seen in Supplementary Movie 1, which shows that the ROS only occurred in the Pt site bonded with CO, whereas the other Pt sites stayed intact. This explains why the ROS process only occurred when CO was introduced, which is in accordance with the results of in situ NAP-XPS and transient CO oxidation studies (Figs. 2e, 3a). As for Pt/TiO$_2$-R (Fig. 5d and Supplementary Movie 2), the atomic distance of Pt-O quickly increased in the first 6 ps, and then stayed at 3-4 Å to the end of AIMD simulation, whereas the atomic distances of the two Ti-O bonds in Ti-O-Ti remained unchanged during the whole AIMD simulation. This means cleavage of Pt-O bond, while the Ti-O-Ti kept intact, indicating that there was no occurrence of ROS over Pt/TiO$_2$-R.

The cycles of complete CO oxidation reaction over Pt/Sn$_{0.2}$Ti$_{0.8}$O$_2$ and Pt/TiO$_2$-R were simulated by Vienna Ab initio Simulation Package (VASP), and the oxidation routes were based on the AIMD simulation, which were named as CO oxidation by ROS (configurations I-VIII in Fig. 5e) and CO oxidation by O$_{latt}$ (configurations i–vii in Fig. 5e). Additionally, the route of CO oxidation by O$_{latt}$ over Pt/Sn$_{0.2}$Ti$_{0.8}$O$_2$ was presented as a reference. It should be noted that these reaction routes followed the MvK mechanism, and other reaction mechanisms were not considered because O$_2$ could not be adsorbed at or around the Pt clusters of Pt/Sn$_{0.2}$Ti$_{0.8}$O$_2$ and Pt/TiO$_2$-R (Supplementary Fig. S23). As for the route of CO oxidation by O$_{latt}$, CO was first adsorbed on the Pt site (i→ii). Then, CO was oxidized by the O$_{latt}$ near Pt to generate adsorbed CO$_2$ (ii→ts-1→iii). With the subsequent desorption of surface CO$_2$, an oxygen vacancy (V$_O$) was left on the surface (iii→iv). The V$_O$ was filled by an oxygen molecule (iv→v). Another CO molecule was adsorbed on the Pt site near the adsorbed O$_2$ (v→vi), and subsequently oxidized by the O atom of O$_2$ to generate surface CO$_2$ (vi→ts-2→vii). Finally, the surface CO$_2$ was released, and the catalyst surface was restored to its original state (vii→i). The energy barrier of CO oxidation by O$_{latt}$ over Pt/Sn$_{0.2}$Ti$_{0.8}$O$_2$ was 0.90 eV, which was slightly higher than that over Pt/TiO$_2$-R (0.84 eV). Such result was inconsistent with the

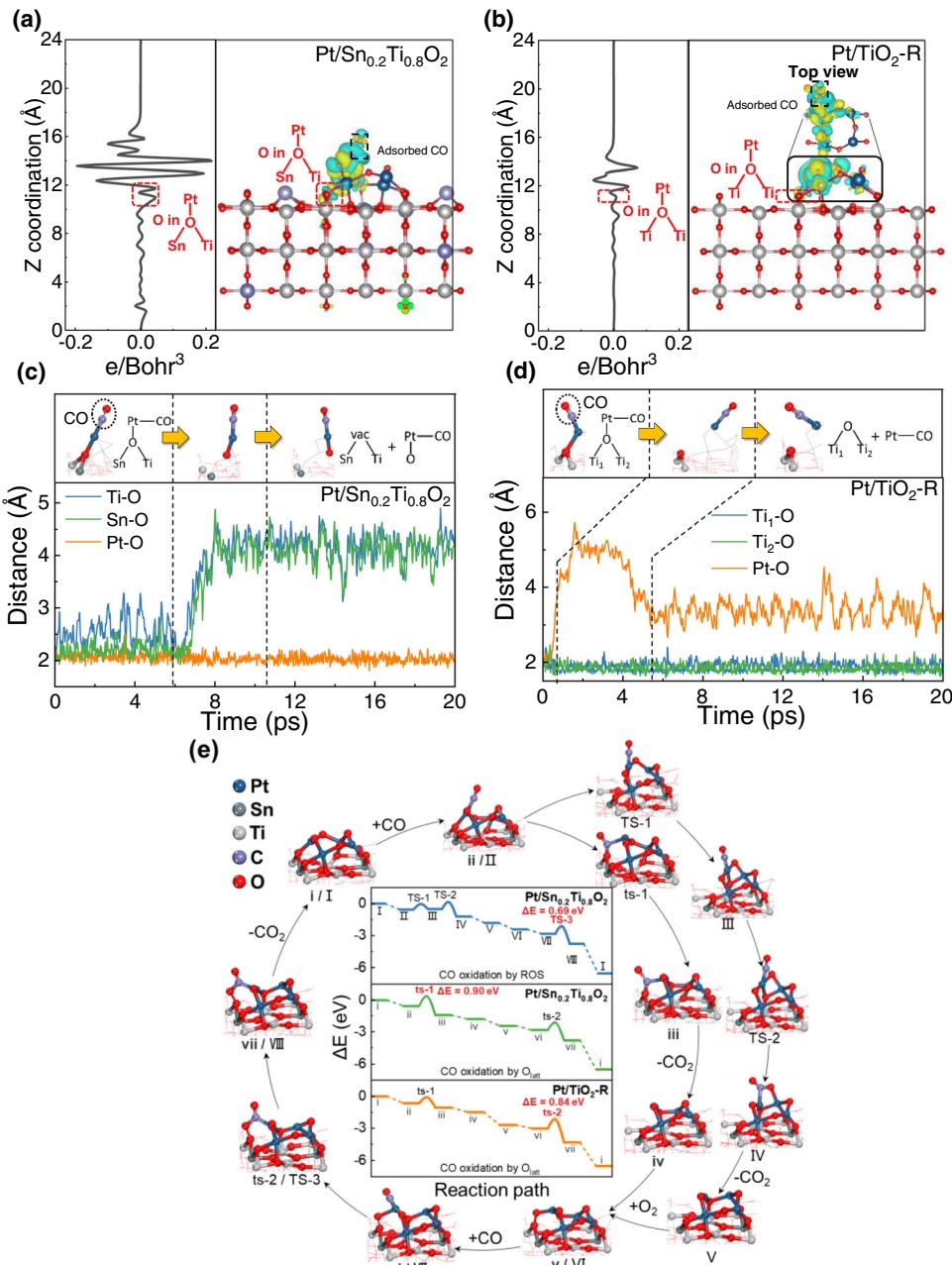

**Fig. 5 | DFT simulations of CO oxidation over Pt/Sn$_{0.2}$Ti$_{0.8}$O$_2$ and Pt/TiO$_2$-R.** **a**, **b** Charge density difference and the corresponding planar-average charge density analyses of CO adsorption on (**a**) Pt/Sn$_{0.2}$Ti$_{0.8}$O$_2$, and (**b**) Pt/TiO$_2$-R. The charge density difference was calculated by the equation $\Delta\rho = \rho_{AB} - \rho_A - \rho_B$, where A represents CO, and B represents Pt/Sn$_{0.2}$Ti$_{0.8}$O$_2$ (**a**) or Pt/TiO$_2$-R (**b**) (Yellow represents an increase of electron density, and blue represents a decrease of electron density). **c**, **d** Atomic distance of Ti-O, Sn-O, and Pt-O as a function of simulated time during AIMD simulation over (**c**) Pt/Sn$_{0.2}$Ti$_{0.8}$O$_2$ and (**d**) Pt/TiO$_2$-R. **e** Energy profiles and configurations of CO oxidation reaction following the pathway involving ROS (I–VIII) and that involving O$_{latt}$ (i–vii).

activity results (Fig. 1a, b), indicating that CO should be oxidized following a different reaction route over Pt/Sn$_{0.2}$Ti$_{0.8}$O$_2$. As for the route of CO oxidation by the O comes from the ROS process, CO was also adsorbed on the Pt site (I→II). Then, the Sn-O and Ti-O bonds in Sn-O-Ti near the Pt-CO site were broken, and the O atom migrated to the Pt clusters (II→TS-1 → III, i.e. ROS process) and left an V$_O$ next to Sn and Ti on the support. Then the adsorbed CO was oxidized by the O in Pt cluster to generate adsorbed CO$_2$ (III → TS-2 → IV). With the desorption of CO$_2$, the residual V$_O$ next to Pt was filled by an O atom transferred through ROS (IV→V). The O atom transferred through ROS could oxidize CO following another reaction cycle. The V$_O$ next to Sn and Ti on the support was filled by an O$_2$ molecule (V→VI), and the subsequent reaction route (VI→VII→TS-3 → VIII→I) was the same as that of CO

oxidation by O$_{latt}$ (v→vi→ts-2→vii→i). The results of this DFT simulation show that the energy barrier of CO oxidation by ROS over Pt/Sn$_{0.2}$Ti$_{0.8}$O$_2$ was 0.69 eV, which was lower than that of CO oxidation by O$_{latt}$ (0.90 eV), suggesting that CO oxidation by the O atom transferred through ROS is probably more preferred over Pt/Sn$_{0.2}$Ti$_{0.8}$O$_2$. In addition, the energy barrier of CO oxidation by ROS over Pt/Sn$_{0.2}$Ti$_{0.8}$O$_2$ (0.69 eV) was lower than that of CO oxidation by O$_{latt}$ over Pt/TiO$_2$-R (0.84 eV), indicating that CO was more easily oxidized by Pt/Sn$_{0.2}$Ti$_{0.8}$O$_2$, which was in accord with the activity results (Fig. 1a, b).

## Discussion

In this study, Sn was doped into TiO$_2$ to induce ROS in Pt/Sn$_{0.2}$Ti$_{0.8}$O$_2$. With the Pt/TiO$_2$-R and Pt/TiO$_2$-A catalysts as references, the Pt/

$Sn_{0.2}Ti_{0.8}O_2$ catalyst exhibited much higher CO catalytic oxidation activity, demonstrating that low-temperature CO oxidation on Pt/$Sn_{0.2}Ti_{0.8}O_2$ was energetically more favorable via the ROS route.

The reaction orders of CO and $O_2$ over Pt/$Sn_{0.2}Ti_{0.8}O_2$, Pt/$TiO_2$-R, and Pt/$TiO_2$-A were all slightly higher than 0, which pointed to the MvK mechanism, typical of catalysts based on reducible oxides. This result was further verified by DFT simulation, which suggests that $O_2$ could not be directly adsorbed at or around the Pt clusters of Pt/$Sn_{0.2}Ti_{0.8}O_2$ and Pt/$TiO_2$-R. Accordingly, the reactivity of the active lattice oxygen was studied by a transient CO oxidation without the presence of gaseous $O_2$. The results reveal that Pt/$Sn_{0.2}Ti_{0.8}O_2$ contained a higher amount of active lattice oxygen, but it cannot be characterized by $O_2$-TPD. Therefore, the idea of "in situ generation of active O species upon CO introduction" over Pt/$Sn_{0.2}Ti_{0.8}O_2$ was conceived.

The assumption of CO-induced ROS was systematically investigated by a series of in situ studies. The in situ NAP-XPS spectra suggest that $Pt^{2+}$ was the major Pt species of Pt/$Sn_{0.2}Ti_{0.8}O_2$, Pt/$TiO_2$-R, and Pt/$TiO_2$-A after the moderate reduction, but only on Pt/$Sn_{0.2}Ti_{0.8}O_2$ catalyst that the generation of $Pt^{4+}$ species was observed when it was exposed to CO or CO + $O_2$. Three possibilities may lead to the formation of $Pt^{4+}$ on Pt/$Sn_{0.2}Ti_{0.8}O_2$: (i) $Pt^{2+}$ captures gaseous $O_2$ thus being oxidized into $Pt^{4+}$; (ii) $Pt^{2+}$ donates an electron to the support, therefore resulting in $Pt^{4+}$ formation; (iii) lattice O in the support migrates to the vicinity of $Pt^{2+}$ (i.e. the ROS process), causing the oxidation of $Pt^{2+}$ to $Pt^{4+}$. In situ NAP-XPS results (Fig. 3a) indicate that $Pt^{4+}$ was not observed when Pt/$Sn_{0.2}Ti_{0.8}O_2$ was exposed to $O_2$ alone, eliminating the first possibility. Regarding the second possibility, it is well-known that Pt clusters supported on stoichiometric $TiO_2$ donate electrons to the support, and the Pt clusters draw electrons from the reduced $TiO_2$. Therefore, when reducing gas CO is introduced, Pt should receive electrons from the support, resulting in lower Pt valence. However, NAP-XPS (Fig. 3a) shows that when reducing gas CO was introduced, $Pt^{2+}$ on the surface of Pt/$Sn_{0.2}Ti_{0.8}O_2$ is oxidized to $Pt^{4+}$, indicating that electron transfer between Pt and the support cannot result in the $Pt^{4+}$ formation. Therefore, the ROS process induced by CO adsorption should be the cause of the oxidation of $Pt^{2+}$ to $Pt^{4+}$. In situ Raman studies showed that there was a weakening of O-Ti(Sn)-O bond strength, demonstrating higher mobility of surface $O_{latt}$ when CO was introduced onto Pt/$Sn_{0.2}Ti_{0.8}O_2$. The in situ DRIFTS spectra indicated that the average valence state of Pt over Pt/$Sn_{0.2}Ti_{0.8}O_2$ obviously increases with rising reaction temperature. The result was further verified by the in situ acquired NAP-XPS spectra conducted under reaction conditions. Such results reveal that ROS over Pt/$Sn_{0.2}Ti_{0.8}O_2$ became more significant with increasing CO oxidation rate. Overall, the doped Sn promoted the mobilities of $O_{latt}$ in support, facilitating the ROS process upon CO introduction. This conclusion substantiates the "in situ generation of active O species" assumption over the Pt/$Sn_{0.2}Ti_{0.8}O_2$ catalyst.

The fundamental details of ROS process were further investigated by DFT simulations. Structure optimization suggests that the $Pt_4O_4$ clusters tended to connect with Sn-O-Ti through the O site in Sn-O-Ti. The charge density of O in these Sn-O-Ti bonds increased after the loading of $Pt_4O_4$ clusters as well as the subsequent adsorption of CO. We speculated that the increased charge density promoted the mobility of the O in Sn-O-Ti, resulting in the occurrence of ROS in Pt/$Sn_{0.2}Ti_{0.8}O_2$ during CO oxidation reaction. Reaction cycle simulations show that over Pt/$Sn_{0.2}Ti_{0.8}O_2$, the energy barrier of CO oxidation directly by lattice O was higher than that by the O species derived from the ROS process, suggesting that the latter is more preferred for CO oxidation over Pt/$Sn_{0.2}Ti_{0.8}O_2$ as depicted in Fig. 6. A detailed visible depiction of the ROS process can be found in the Supplementary Movies 1 and 2 of AIMD simulations.

Overall, we activated the low-temperature reverse oxygen spillover on Titania-supported Platinum catalyst by introducing Sn into $TiO_2$ support, and further demonstrated the existence and mechanistic route of reverse oxygen spillover in low-temperature CO oxidation by

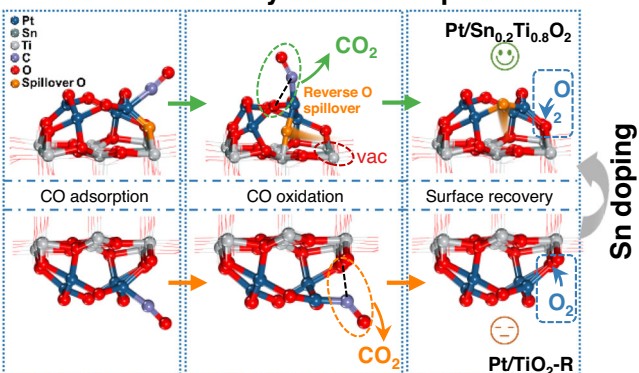

**Fig. 6 | Schematic of CO oxidation on Pt/TiO₂ catalysts with and without reverse oxygen spillover.** The upper part show CO is oxidized by reverse spillover O after Sn doping, and the lower part show CO is oxidized by lattice O without Sn doping.

a combination of experimental and theoretical studies. The revealed interfacial dynamics in reverse oxygen spillover fills the gaps of interfacial chemistry of adsorbates and/or intermediate transport through metal support interfaces, and allows deeper fundamental understanding of catalytic reactions involving oxygen, and hence improvement of catalyst design for technologically relevant redox reactions. We speculate the strategy to improve the oxygen mobility of support, such as the construction of asymmetric oxygens ($M_1$-O-$M_2$) or doping secondary metals with weak M-O bond, probably is particularly effective to promote ROS. Looking forward, we anticipate that the reactant-adsorption-triggered characteristics of oxygen spillover will arouse further investigations on the mechanistic effect of microstructures (such as size of active sites, nature of supports, degree of O charging) and reactants (such as hydrocarbons, $H_2O$ and $SO_2$), finding more potential applications in thermo-, photo- and electrocatalysis.

## Methods
### Synthesis of materials
The $Sn_xTi_{1-x}O_2$ supports (where x represents the molar ratio of Sn/(Sn +Ti)) were synthesized by a co-precipitation method using $SnCl_4\cdot5H_2O$ and $Ti(SO_4)_2$ as precursors[57]. First, $SnCl_4\cdot5H_2O$ and $Ti(SO_4)_2$ were dissolved in deionized water. Then, a standard ammonia solution (25 wt%) was added into the solution until pH 10 to induce the co-precipitation of Sn and Ti ions. The precipitates were filtered out and washed by deionized water until neutral. The obtained materials were dried at 105 °C for 12 h and subsequently calcined at 500 °C for 4 h in air with a heating rate of 2 °C/min. The reference anatase $TiO_2$ was prepared by the same method but without $SnCl_4\cdot5H_2O$, and was denoted as $TiO_2$-A due to its anatase crystallite. A rutile $TiO_2$ (Aladdin, 99.99%) was directly employed as another reference sample, and was denoted as $TiO_2$-R.

All the Pt/$Sn_xTi_{1-x}O_2$ and Pt/$TiO_2$ catalysts were prepared by the impregnation method using platinum nitrate (Pt($NO_3$)$_2$) solution as metal source. First, commercial solution of Pt($NO_3$)$_2$ (Aladdin, 18.02% of Pt) was diluted to 0.01 g Pt/mL. Then 1.0 mL of the diluted Pt($NO_3$)$_2$ solution was added into 2 g of support ($TiO_2$ or $Sn_xTi_{1-x}O_2$) with vigorous stirring at room temperature. The obtained samples of 0.5 wt% Pt loading were dried at 105 °C for 12 h, and then calcined at 500 °C for 1 h in air with a heating rate of 2 °C/min. Finally, the Pt/$Sn_xTi_{1-x}O_2$ and Pt/$TiO_2$ catalysts were treated in 5% $H_2/N_2$ at 300 °C for 1 h.

### CO oxidation performance
The performance of catalysts (100 mg, 40–60 mesh) in CO oxidation was evaluated using a fixed-bed quartz micro-reactor. The typical

reaction condition was as follows: 1% CO, 1% $O_2$, and $N_2$ as balance gas with a total flow rate of 100 mL min$^{-1}$, corresponding to a gas hourly space velocity (GHSV) of 60,000 mL $g_{cat}^{-1}$ h$^{-1}$. The concentrations of CO and $CO_2$ in the inlet and outlet streams were measured by an infrared gas analyzer (Gasmet Dx-4000). The CO conversion was calculated based on the following equation:

$$X_{CO} = \frac{C_{CO_{in}} - C_{CO_{out}}}{C_{CO_{in}}} \times 100\% \qquad (1)$$

where, $X_{CO}$ is CO conversion, $C_{COin}$, and $C_{COout}$ are the concentrations of CO in the inlet and outlet.

The specific reaction rates and TOF of CO oxidation at different temperatures over the catalysts were measured under different conditions, keeping CO conversion below 20% by varying the GHSV (to suppress the influence of inner and external diffusion).

The reaction rate k (µmol g$^{-1}$ s$^{-1}$) can be calculated by assuming ideal gas behavior:

$$k = \frac{X_{CO} \cdot F_{CO}}{W} \times 100\% \qquad (2)$$

where, $X_{CO}$ is CO conversion, $F_{CO}$ (µmol s$^{-1}$) is molar gas flow rate of CO, and $W$ (g) is the mass of catalyst in the fixed-bed reactor.

The catalytic velocities were determined by a turnover frequency of CO conversion over Pt sites (TOF$_{Pt}$ (s$^{-1}$)), which can be obtained by the following equation:

$$TOF_{Pt} = \frac{X_{CO} \cdot F_{CO}}{N_{Pt}} \times 100\% \qquad (3)$$

where, $X_{CO}$ is CO conversion (<20%), $F_{CO}$ (µmol s$^{-1}$) is molar gas flow rate of CO, and $N_{Pt}$ (µmol) is the total number of Pt atoms on the catalyst. It should be noted that all the catalysts used in this study contained 0.5 wt% Pt. Moreover, all Pt atoms were considered when calculating TOF$_{Pt}$ to enable a fair comparison of catalyst activity under the same Pt usage.

The apparent activation energies ($E_a$ (kJ mol$^{-1}$)) over the catalysts were calculated at CO conversion lower than 20% according to the Arrhenius equation.

## Characterization

Transient CO oxidation was tested in a fixed-bed quartz micro-reactor. The reduced catalysts were loaded into the reactor and heated to the reaction temperature under $N_2$ without any further pretreatment. The reaction temperature was fixed at 100 or 200 °C with a CO concentration of 1% and $N_2$ as the balance gas, without the supply of $O_2$. The gas flow rate was set to 100 mL min$^{-1}$ with a GHSV of 60,000 mL $g_{cat}^{-1}$ h$^{-1}$. The infrared gas analyzer (Gasmet Dx-4000) was utilized to measure the concentrations of CO and $CO_2$ in both the inlet and outlet streams.

$N_2$ physisorption was measured at liquid nitrogen temperature by a micromeritics ASAP 2460 instrument in the static mode. Before measurement, the catalysts were degassed at 250 °C for 4 h. The specific surface area was calculated by the Brunauer-Emmett-Teller (BET) equation. The pore volumes and average pore diameters were determined using the Barrett-Joyner-Halenda (BJH) method based on the $N_2$ adsorption-desorption isotherms.

XRD patterns were recorded over D8 Advance X-ray diffractometer (Bruker AXS company) using Cu K$_\alpha$ radiation. The pattern was recorded in the 2θ range of 5–90° with a speed of 1.5°/min and a step size of 0.02°. The operating voltage and current was 40 kV and 40 mA, respectively. Rietveld refinements of all XRD patterns were performed by the GSAS software package.

$O_2$-TPD study was conducted on a chemisorption analyzer (Micromeritics, AutoChem 2920 ThermoStar). In a quartz reactor, the

catalyst sample was pretreated at 300 °C in 2% $O_2$/He for 1 h. Then, the catalyst was cooled to 50 °C in a flow of 2% $O_2$/He for 30 min. Then the weakly adsorbed O species were purged by pure He for 30 min, and the temperature was subsequently increased to 800 °C with continuous introduction of He at a rate of 10 °C min$^{-1}$. The desorption of $O_2$ was recorded using a thermal conductivity detector (TCD).

SEM and TEM images were observed by ZEISS GEMINISEM 500 electron field emission scanning electron microscope at 30 kV, and JEM 2100 F microscope operating at 200 kV, respectively.

In situ NAP-XPS spectra were recorded by a SPECS-AU190069 instrument. The instrument was equipped with an advanced multi-stage differential pumping system and static voltage lens, which can be used in ultra-high vacuum ($1 \times 10^{-9}$ mbar) with gases of 0–5 mbar. All spectra were collected by using monochromatized Al Kα irradiation (1486.6 eV), which was generated by 50 W of excitation source power in an Al anode (SPECS XR-50). The generated X-ray spot was -0.3 mm in diameter, close to the aperture of the nozzle. The reaction pressure was kept at 1 mbar by a pressure-reducing valve. The powder sample was pressed into a smooth sheet, and was fixed on a special sample table that can be heated during reaction. An electron flood gun was equipped to compensate the charging of catalysts during measurements.

In situ DRIFTS study was carried out on a Nicolet iS50 FTIR spectrometer equipped with a DRIFTS cell and a highly sensitive MCT detector. The DRIFTS spectra were collected by accumulating 64 scans with a spectral resolution of 4 cm$^{-1}$. To exclude the influence of moisture and impurities, the catalyst was pretreated at 300 °C in 5%$H_2$/He for 1 h before each test, then purged with pure He for 1 h. After the pretreatment, the catalyst was cooled to a designated temperature. The spectrum was collected after the introduction of 1% CO and 1% $O_2$ with $N_2$ balance for 30 min.

In situ Raman study was conducted by a Horiba LabRAM HR Evolution instrument equipped with 532 nm laser source (Ventus LP 532), Synapse Charge Coupled Device (CCD) detector, and in situ cell reactor (Linkam CCR1000). The catalyst was pretreated at 300 °C in 5% $H_2$/$N_2$ for 1 h before each test, then purged with pure $N_2$ for 1 h. After the pretreatment, the catalyst was cooled to 100 °C for spectrum collection. The Raman spectra were collected by accumulating eight scans with an acquisition time of 6 s. Then, 1% CO and/or 1% $O_2$ with $N_2$ balance were introduced, and the corresponding Raman spectra were recorded after 30 min of reaction.

SAC-STEM image was observed by using a JEM ARM200F transmission electron microscope operating at 200 kV, which was equipped with a probe corrector, a high-angle annular dark-field detector, and a EDX detector.

XANES and EXAFS were conducted at Singapore Synchrotron Light Sources. The radiation was monochromatized by a Si (111) double crystal monochromator, and the results were processed by employing the Athena software.

## DFT simulation

The slab models were designed rationally. First, because the HR-TEM images indicate that the (110) plane of rutile was the main exposed plane of Pt/Sn$_{0.2}$Ti$_{0.8}$O$_2$ and Pt/TiO$_2$-R, we employed the (110) plane of rutile TiO$_2$ as substrate with a [5 × 3] supercell and three stoichiometric TiO$_2$ layers (-15 Å × 20 Å × 9 Å). Second, since the SAC-STEM images and XPS results suggest that the Pt species on the support was in the form of PtO nano clusters, a Pt$_4$O$_4$ cluster was placed on this (110) plane of rutile TiO$_2$ to simulate the configuration of Pt/TiO$_2$-R. Then, 20% of Ti atoms in the configuration of Pt/TiO$_2$-R was randomly replaced by Sn atoms to simulate the configuration of Pt/Sn$_{0.2}$Ti$_{0.8}$O$_2$. All these slab models were separated by a vacuum of 15 Å.

AIMD simulation was performed using the CP2K package. The generalized-gradient approximation (GGA) with spin-polarized Perdew–Burke–Ernzerh (PBE) functional was used to describe the exchange-correlation energy. A double-z Gaussian basis sets with an

auxiliary plane wave basis set (cutoff energy of 500 Rydberg) was used to expand the wavefunctions. All AIMD simulations were conducted by sampling the canonical ensemble with Nose–Hoover thermostats and a time step of 1 fs. Because of the relatively short time scale of AIMD (10–20 ps in this study), the sampling of AIMD was only applied to extremely fast events of low-energy barrier, and the observation of slow processes was excluded. Hence a large number of phase spaces could be explored quickly by enhancing the simulation temperature, and thus the AIMD simulation temperature was set at 700 K, slightly lower than the temperature adopted for the calcination of the samples[56]. The configurations of $Pt/Sn_{0.2}Ti_{0.8}O_2$ and $Pt/TiO_2$-R first underwent an AIMD simulation of 10 ps to ensure the stabilization of these configurations. Then, the configurations at 2, 4, 6, 8, and 10 ps of AIMD simulation were set as initial structure and optimized by VASP to obtain the most stable configurations of $Pt/Sn_{0.2}Ti_{0.8}O_2$ and $Pt/TiO_2$-R. Finally, a CO molecule was adsorbed on the configurations of $Pt/Sn_{0.2}Ti_{0.8}O_2$ and $Pt/TiO_2$-R to undergo an AIMD simulation of 20 ps to investigate the reverse O spillover during CO oxidation reaction.

The geometrical optimization and transition state (TS) retrievals were performed by VASP with projected augmented wave (PAW). The PBE functional was used in the GGA with the Hubbard model, which was expanded on a plane wave basis with 400 eV of kinetic cutoff energy. The $U_{eff}$ (i.e., $U$–$J$) of Ti[58], Sn[59], and Pt[60–62] were set at 4.5, 5.0, and 9.0 eV, respectively. The gamma point (i.e., $1 \times 1 \times 1$) of the K-point was employed due to the large sizes of these configurations. The SCF tolerance and maximum atomic force were set at $10^{-6}$ eV and 0.02 eV/Å during geometrical optimization. The transition state was roughly estimated by the CI-NEB (climbing the image-nudged elastic band) method. Then the roughly converged transition state was precisely optimized by the Dimer method.

## Data availability
The data that support the findings of this study are included in the published article (and its Supplementary Information) or available from the corresponding author upon reasonable request.

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

## Acknowledgements

This research was supported by the National Key R&D Program of China (no. 2022YFC3701600), Fundamental Research Funds for the Central Universities of China (Nos. 2682022CX035 and 2682022KJ035), National Natural Science Foundation of China (nos. 52070114 and 22206155), Sichuan Science and Technology Program (no. 2023JDRC0066), China Postdoctoral Science Foundation (no. 2022M712632) and State Environmental Protection Key Laboratory of Sources and Control of Air Pollution Complex (no. SCAPC202109). Dr. Xuefeng Chu and Xiaoping Chen are acknowledged for the XPS characterizations. Xiaoping Chen is acknowledged for the design of TOC.

## Author contributions

J.J.C. and S.C.X. proposed the idea and wrote the paper. J.J.C. and J.H.Li. supervised the whole work. S.C.X. performed a DFT simulation. H.Y.L. and J.Q.S. performed all the experiments except in situ DRIFTs. L.O. and F.M. designed the in situ DRIFTs and modified the manuscript. J.X.M., H.L., and Z.J.G. modified the manuscript. All the authors discussed the results and commented on the manuscript.

## Competing interests

The authors declare no competing interests.
