## [Peer Review File · Nature Communications]

Reverse Oxygen Spillover Triggered by CO Adsorption on Titania-Supported Platinum Catalyst for the Boosting of Low-Temperature CO OxidationREVIEWER COMMENTS

Reviewer #1 (Remarks to the Author):

In the manuscript by J. Chen et al., the experimental and theoretical results on the enhanced catalytic activity on Sn-doped Pt/TiO₂ catalysts that were associated with the low temperature reverse oxygen spillover. A combination of near-ambient-pressure XPS, in situ Raman/Infrared spectroscopies, and ab initio molecular dynamics (AIMD) simulations revealed that the ROS was triggered by CO adsorption at Pt 2+ sites, followed by bond cleavage of Ti-O-Sn moieties nearby and the appearance of Pt⁴⁺ species. I find the result addresses the role of the interfacial chemistry of ROS that is triggered by CO adsorption, which is an important subject in heterogeneous catalysis. However, there are many points that need to be clarified, also additional experimental results should be presented to support the key claim of the paper. Below are my detailed comments that need to be addressed.

1. The smaller Pt nanoparticles exhibit higher oxidation states [as addressed earlier, For example, Somorjai and Park *Angewandte Chemie Int. Ed.* 47, 9212 (2008)] However, I think the Pt nanoparticles with the size of 1-2 nm should exhibit some metallic peaks, which were absent in the current investigation. I think the authors need to mention the earlier studies on the size dependence of Pt oxidation states and discuss why metallic Pt peaks are not visible.
2. The experimental conditions for various operando experiments should be described more precisely. In near ambient pressure XPS presented in Fig 3 and Figure 4, what is the partial pressure during the ambient pressure experiment? For example, I assume that the authors measured Figure 3a under ambient pressure. However, the figure caption indicated that XPS data of Pt 4f spectra after the sample was exposed to O₂, CO+O₂, and CO at 100 C. The way it is presented is so confusing. The partial pressure and the types of gas during the near-ambient-pressure XPS, in situ Raman/Infrared spectroscopies, should be more clearly indicated.
3. I think the temperature dependence of XPS measurement under CO and CO+O₂ should be addressed in order to claim the low temperature reverse oxygen spillover. I am wondering if the authors can measure the XPS of rPt/Sn_{0.2}Ti_{0.8}O₂ catalysts under gas conditions by changing the temperature to verify the evolution of Pt²⁺ and Pt⁴⁺ peaks.
4. It is hard to follow the way of XPS deconvolution. It was mentioned that the peaks at ~72.2 eV and ~74.8 eV could be assigned to 4f_{7/2} and 4f_{5/2} signals of Pt 2+ species, whereas the peaks at ~74.4 eV and ~77.8 eV to 4f_{7/2} and 4f_{5/2} signals of Pt 4+ species, respectively. These two peaks of 4f_{7/2} and 4f_{5/2} signals should have a similar number of FWHM (full width of half maximum), but they are randomly changing (see Fig. S3, etc.). This point needs to be clarified.

Reviewer #2 (Remarks to the Author):

This manuscript by Li and coworkers address the mechanism of CO-induced oxygen spillover from the Sn-doped TiO₂ to supported Pt NPs using several in-situ techniques and DFT-based theoretical analyses. I think this is a comprehensive work and publishable. However, this manuscript does not satisfy the quality criteria for publication in Nature Communications. The bottom line is that the oxygen chemistry at the Pt-TiO₂ interfaces varies as Sn dopants are introduced. However, some vague experimental points make the authors' statements controversial.

Spillover of oxygen species from the supporting oxides to the supported metal NPs has always been considered in studying the catalytic functionality of the metal-oxide interfaces. Even in the cases of several early studies of CO oxidation catalyzed by oxide-supported metal NPs, reverse oxygen spillover has been considered a possible oxygen-supplying pathway. Moreover, the mechanism reported here by the authors is conceptually equivalent to oxygen taking over, which usually happens during the Mars-van Krevelen-type CO oxidation processes. I understand that the conceptual difference between the reverse oxygen spillover claimed by the authors with a conventional MvK type mechanism is whether the spillover oxygen can lay stable on the surface of Pt clusters or nanoparticles.

I am skeptical whether the presence of Pt⁴⁺ directly confirms that oxygen was moved from the Sn-doped TiO₂ to Pt clusters. It is well-known that Pt clusters supported on a stoichiometric TiO₂ donate electrons to the support and that the Pt clusters draw electrons from the reduced TiO₂. Moreover, although the oxygen vacancy appeared to be associated with oxygen spillover, bulk oxygen can heal this vacancy thoroughly. If the local coordinative ensemble at the Pt-TiO₂ interfaces varies dynamically, I believe that the concentration of the Pt⁴⁺-like species may accordingly vary.

I am also not sure whether the surface of sub-nanometer- or nanometer (<2nm)-sized Pt clusters (experimentally observed ones) competitively binds oxygen rather than CO. This is presumably the key to whether the authors' claim can be rationalized or not. I am sure that the small Pt clusters strongly bind CO more than oxygen. In the cases of the larger Pt nanoparticles (than 2nm) in which micro-facets can develop at their surfaces, oxygen may share some portion of the surface with CO. However, it is hard to agree with the authors' claim that their small Pt clusters grip and stabilize oxygen at their surface. I agree that such oxygen can limitedly appear in the early stage of the reaction to which the catalyst is exposed to oxygen without CO. However, such oxygen species can be easily removed by CO (as presented in Fig. 5), and CO will preferentially occupy the surface of the Pt clusters. I am unsure whether an oxygen molecule heals the open position at stage v/VI (Fig. 5).

Moreover, I think assigning the PtO and Pt-O-Ti separately in XAS, or XPS spectra is risky because the local atomic ensemble at the metal-oxide is complicated, and dynamic electronic variation occurs during the reaction (as reported by Daelman et al. Nat. Mater. 2019,18, 1215).

Fig. 5 shows the overall CO oxidation pathway following the Langmuir-Hinshellwood mechanism. This is, unfortunately, equivalent to the case of CO oxidation by Pt-oxide clusters, as an oxygen molecule is required to restore the initial Pt-oxide cluster.

On page 16, “The energy barrier of CO oxidation by ROS over Pt/Sn_{0.2}Ti_{0.8}O₂ was 0.69 eV, which was obviously lower than that of CO oxidation by O_{latt} (0.90 eV), suggesting that CO oxidation by the O atom transferred through ROS is more preferred over Pt/Sn_{0.2}Ti_{0.8}O₂.” I think this kind of statement made from the DFT calculations on flat TiO₂ surfaces cannot be generalized. The barrier for the direct MvK mechanism will decrease and become sensitive to the surface morphology of TiO₂, which is the case of the experimentally synthesized catalyst.

Reviewer #3 (Remarks to the Author):

In this work, the authors synthesized Pt-based catalysts supported on Sn_xTi_{1-x}O₂, TiO₂(anatase) and TiO₂(rutile) supports. Catalyst structures were thoroughly characterized by electron microscopy, XRD, AP-XPS, DRIFTS and in-situ RAMAN. The catalytic activity was probed by CO oxidation, and it was found that Pt/Sn_xTi_{1-x}O₂ exhibited improved low temperature CO oxidation than Pt/TiO₂(anatase and Pt/TiO₂(rutile). Therefore, the authors devoted extended efforts to understand the nature of such improved low temperature activity promotion when Sn was used to dope TiO₂. Overall, Pt particle size was similar across all samples and CO-TPR experiments revealed improved lattice oxygen availability on Pt/Sn_xTi_{1-x}O₂. AP-XPS and in-situ RAMAN were critical to identify that upon the exposure of Pt/Sn_xTi_{1-x}O₂ to CO+O₂, increased Pt⁴⁺ species and weaker Ti-O-Sn bonds were detected, which suggests lattice oxygen from Ti-O-Sn could be mobilized towards Pt centers (process called Reverse Oxygen Spillover by the authors (ROS)). Computational chemistry calculations (AIMD) illustrated that ROS occurs on model Pt₄O₄/Ti-O-Sn and that lowest energy barrier CO oxidation pathway occurs via ROS rather than other routes.

I recommend this article for publications only if the authors can satisfactorily address the following concerns, clarifications, and improvements to the manuscript.

1. Novelty

Q1.1. Reverse spill over of oxygen has recently been reported in JACS (J. Am. Chem. Soc. 2023, 145, 2523–2531). Such publication is crucially relevant for the present manuscript and should be included in the introduction. It also raises the question on the novelty of this manuscript. Authors should clearly clarify the novelty of this manuscript or what additional knowledge or novelty this manuscript brings over what already been reported in the JACS report.

Q1.2. The authors should provide clear rationales on why Sn was selected to dope TiO₂ instead of any other promoters. What are the reasons/hypothesis/preliminary work behind such selection? What is the generality of selecting a promoter such as Sn to achieve such performances.

2. Clarifications

Q2.1. Line 86. The author should use the term USDRIVE, not USDRIVER. (Please see the following reference: <https://www.energy.gov/eere/vehicles/us-drive>)

Q2.2. How were TOFs estimated? I could not find how the fraction of active Pt species was estimated for the TOF calculations. Please clarify and provide a detailed explanation of methods.

Q3.3. Include detailed CO-TPR (shown in Figure 2e) experiment description in methods section. To assess the validity of author's claim about correlation between active species in the CO-TPR and O₂-TPD experiments (shown in figure 2e and 2f) it is crucial for reviewers to understand exactly how the CO-TPR

was carried out, including any pre-treatments with O₂ or purge of weakly adsorbed O₂ species. Since the main finding of this paper (Reverse Oxygen Spillover on Ti/Sn_xTi_{1-x}O₂) is based CO-TPR and O₂-TPD experiment analysis (lines 222-226), I highly suggest that further clarification of experiments is provided before this work can be considered for publication.

Q3.4. It would be highly appreciated if authors could indicate which Sn-O-Ti bonds are broken in CO oxidation cycle on the ROS route to facilitate understanding of reaction cycle schemes.

3. Improvements

Q3.1. The discussion on why several reduction treatments were done to finally select 300 °C in H₂ is not scientifically interesting/relevant to the main goal of the study (understanding the ROS effect) and should be placed in the SI section.

Q3.2. Indicate error bars in figures S5B and S5C (data set in Arrhenius plots) as well as in all figures containing TOF measurements.

Q3.3. Lines 126. I recommend including a more elaborated explanation on why the observed CO and O₂ partial orders are ~zero. The following work can help with that *Angew.Chem.Int.Ed.*2021,60,26054–26062.

Q3.4. Lines 137-145. Is the observed sulfur tolerance expected for Pt- and TiO₂-based materials? If SO₂-tolerance is not novel due to Sn doping (I don't think so), discussion of relevant literature is appropriate. (example: *NATURE CATALYSIS* | VOL 2 | JULY 2019 | 614–622)

Response to referees

Reviewer #1 (Remarks to the Author):

In the manuscript by J. Chen et al., the experimental and theoretical results on the enhanced catalytic activity on Sn-doped Pt/TiO₂ catalysts that were associated with the low temperature reverse oxygen spillover. A combination of near-ambient-pressure XPS, in situ Raman/Infrared spectroscopies, and ab initio molecular dynamics (AIMD) simulations revealed that the ROS was triggered by CO adsorption at Pt²⁺ sites, followed by bond cleavage of Ti-O-Sn moieties nearby and the appearance of Pt⁴⁺ species. I find the result addresses the role of the interfacial chemistry of ROS that is triggered by CO adsorption, which is an important subject in heterogeneous catalysis. However, there are many points that need to be clarified, also additional experimental results should be presented to support the key claim of the paper. Below are my detailed comments that need to be addressed.

Response: Thank you very much for your valuable comments, which allow us to further improve the manuscript. Hereinafter the one-by-one response to each specific comment as well as the revision accordingly.

1. The smaller Pt nanoparticles exhibit higher oxidation states [as addressed earlier, For example, Somorjai and Park *Angewandte Chemie Int. Ed.* 47, 9212 (2008)] However, I think the Pt nanoparticles with the size of 1-2 nm should exhibit some metallic peaks, which were absent in the current investigation. I think the authors need to mention the earlier studies on the size dependence of Pt oxidation states and discuss why metallic Pt peaks are not visible.

Response: Thank you very much for your valuable suggestions. We have carefully read the literature you mentioned and conducted research on some related studies. Previous studies have suggested a correlation between the oxidation state of surface Pt and the nanoparticle size of the catalyst. In general, the smaller the nanoparticle size of Pt clusters, the higher the oxidation state. For instance, research by Somorjai (*Angew. Chem. Int. Edit.* 47, 9212-9228 (2008)) has demonstrated that in Pt₂₀ clusters (~0.8 nm), Pt is primarily present as Pt²⁺, while in Pt₄₀ clusters (~1.5 nm), Pt is predominantly present as Pt⁰. However, the valence state of Pt clusters can also vary depending on the chemical environment, even when the particle size is the same.

Ozturk et al. (*Langmuir* 21, 3998-4006 (2005)) found that Pt 4f_{7/2} peaks were located at 74.6 eV (representing primarily Pt⁴⁺) and 73.3 eV (representing primarily Pt²⁺) for Pt₄₀ nanoparticles deposited as a thick and a thin layer, respectively. Ye et al. (*Langmuir* 20, 2915-2920 (2004).) observed that the Pt 4f_{7/2} peak presented at 73.0 eV for Pt₃₀ nanoparticles, suggesting that in this case, Pt is mainly present in the form of Pt²⁺.

In our study, the SAC-STEM HAADF images (Fig. 2d) revealed an average particle size of Pt nanoclusters on the surface of the Pt/Sn_{0.2}Ti_{0.8}O₂ catalyst of approximately 1.2 nm, which is close to the particle size of Pt₃₀ nanoclusters. Considering the findings of the aforementioned literature, it is acceptable that the dominant valence state of Pt on the surface of the Pt/Sn_{0.2}Ti_{0.8}O₂ catalyst is 2+. To further confirm the stability of Pt²⁺ on the surface of the Pt/Sn_{0.2}Ti_{0.8}O₂ catalyst, we recorded the XPS spectra of Pt 4f after H₂ reduction at 300-500°C (Fig. S2b). The results showed that Pt²⁺ can still exist stably on the surface of the Pt/Sn_{0.2}Ti_{0.8}O₂ catalyst even after H₂ reduction at 500 °C. The aforementioned literature review and discussion of the valence state of Pt in the catalyst of this study have been added to Supplementary Note 2, with details presented as follows:

"Previous studies have suggested a correlation between the valence state of surface Pt and the nanoparticle size of the catalyst. In general, the smaller the nanoparticle size of Pt clusters, the higher the oxidation state. For instance, research by Somorjai^{S1} has demonstrated that in Pt₂₀ clusters (~0.8 nm), Pt is primarily present as Pt²⁺, while in Pt₄₀ clusters (~1.5 nm), Pt is predominantly present as Pt⁰. However, the valence state of Pt clusters can also vary depending on the chemical environment, even when the particle size is the same. Ozturk et al.^{S2} found that Pt 4f_{7/2} peaks were located at 74.6 eV (representing primarily Pt⁴⁺) and 73.3 eV (representing primarily Pt²⁺) for Pt₄₀ nanoparticles deposited as a thick and a thin layer, respectively. Ye et al.^{S3} observed that the Pt 4f_{7/2} peak presented at 73.0 eV for Pt₃₀ nanoparticles, suggesting that in this case, Pt is mainly present in the form of Pt²⁺. Therefore, in this study, XPS spectroscopy was utilized to investigate the oxidation state of surface Pt under different H₂ reduction temperatures and before/after H₂ reduction at 300 °C. As shown in Supplementary Fig. S2b, the Pt/Sn_{0.2}Ti_{0.8}O₂ catalyst surface contained a small amount of Pt⁴⁺ and a large amount of Pt²⁺ before H₂ reduction. After H₂ reduction at 300 °C, Pt⁴⁺ on the surface of Pt/Sn_{0.2}Ti_{0.8}O₂ was reduced to Pt²⁺, and no further reduction from Pt²⁺ to Pt⁰ was observed even at an increased reduction temperature of

500 °C, indicating that Pt²⁺ can remain stable on the surface of Pt/Sn_{0.2}Ti_{0.8}O₂. Furthermore, after H₂ reduction at 300 °C, Pt species on the surface of Pt/Sn_{0.2}Ti_{0.8}O₂, Pt/TiO₂-R, and Pt/TiO₂-A catalysts were all present mainly in the form of Pt²⁺ (Supplementary Fig. S3)."

Fig. 2d. SAC-STEM HAADF images; inset shows the particle size distribution of Pt clusters.

Supplementary Fig. S2b Pt 4f XPS spectra of Pt/Sn_{0.2}Ti_{0.8}O₂ before and after pretreatment with H₂ at different temperatures.

Supplementary Fig. S3 Pt 4f XPS spectra over original and 300 °C H₂-pretreated (a) Pt/Sn_{0.2}Ti_{0.8}O₂, (b) Pt/TiO₂-R and (c) Pt/TiO₂-A.

2. The experimental conditions for various operando experiments should be described more precisely. In near ambient pressure XPS presented in Fig 3 and Figure 4, what is the partial pressure during the ambient pressure experiment? For example, I assume that the authors measured Figure 3a under ambient pressure. However, the figure

caption indicated that XPS data of Pt4f spectra after the sample was exposed to O₂, CO+O₂, and CO at 100 °C. The way it is presented is so confusing.

The partial pressure and the types of gas during the near-ambient-pressure XPS, in situ Raman/Infrared spectroscopies, should be more clearly indicated.

Response: We sincerely apologize for any confusion caused by our previous description. We have revised the titles of Figures 3 and 4 (i.e. the near-ambient-pressure XPS and in situ Raman/Infrared spectroscopies), providing detailed information on the partial pressures of each gas component, testing temperature, pressure, and other relevant conditions. The revised titles of Figures 3 and 4 are as follows:

"Fig. 3 | *In situ* NAP-XPS and *in situ* Raman studies of Pt/Sn_{0.2}Ti_{0.8}O₂, Pt/TiO₂-R and Pt/TiO₂-A pretreated with 5% H₂ at 300 °C. a–c, *In situ* NAP-XPS Pt 4f spectra recorded at 100 °C after (a) Pt/Sn_{0.2}Ti_{0.8}O₂, (b) Pt/TiO₂-R and (c) Pt/TiO₂-A was exposed to 1 mbar of O₂, 0.5 mbar of CO with 0.5 mbar of O₂, and 1 mbar of CO in turn. d–f, *In situ* Raman spectra recorded at 100 °C after (d) Pt/Sn_{0.2}Ti_{0.8}O₂, (e) Pt/TiO₂-R and (f) Pt/TiO₂-A was exposed to 1% O₂, 1% CO + 1% O₂, and 1% CO in turn under ambient pressure.

Fig. 4 | *In situ* DRIFTS and *in situ* NAP-XPS spectra of Pt/Sn_{0.2}Ti_{0.8}O₂, Pt/TiO₂-R and Pt/TiO₂-A recorded at 30–100 °C after exposure to CO+O₂. a–c, *In situ* DRIFTS spectra recorded at 30–100 °C after (a) Pt/Sn_{0.2}Ti_{0.8}O₂, (b) Pt/TiO₂-R and (c) Pt/TiO₂-A was exposed to 1% CO + 1% O₂ at a designated temperature under ambient pressure. d–f, *In situ* NAP-XPS Pt 4f spectra recorded at 30–100 °C after (d) Pt/Sn_{0.2}Ti_{0.8}O₂, (e) Pt/TiO₂-R and (f) Pt/TiO₂-A was exposed to 0.5 mbar of CO with 0.5 mbar of O₂ at a designated temperature."

3. I think the temperature dependence of XPS measurement under CO and CO+O₂ should be addressed in order to claim the low temperature reverse oxygen spillover. I am wondering if the authors can measure the XPS of Pt/Sn_{0.2}Ti_{0.8}O₂ catalysts under gas conditions by changing the temperature to verify the evolution of Pt²⁺ and Pt⁴⁺ peaks.

Response: CO₂ is generated when CO is solely introduced over Pt/Sn_{0.2}Ti_{0.8}O₂ due to the consumption of active oxygen species on the surface (see Fig. 2e). Consequently, the chemical state of Pt on the surface of Pt/Sn_{0.2}Ti_{0.8}O₂ catalyst is significantly affected by the prolonged introduction of CO alone, which has been demonstrated by

in situ DRIFTS analysis, as depicted in Fig. 1 (for review only). The characteristic peak representing $\text{Pt}^{\delta+}\text{-CO}$ in the in situ DRIFTS spectra of $\text{Pt}/\text{Sn}_{0.2}\text{Ti}_{0.8}\text{O}_2$ shifts to lower wavenumbers as the temperature rises during the introduction of CO alone. In contrast, the characteristic peak representing $\text{Pt}^{\delta+}\text{-CO}$ in $\text{Pt}/\text{Sn}_{0.2}\text{Ti}_{0.8}\text{O}_2$ shifts to higher wavenumbers as the temperature rises during the introduction of CO+O₂ (Fig. 4a). The characteristic peak of $\text{Pt}^{\delta+}\text{-CO}$ is sensitive to various factors, such as Pt dispersion, Pt microstructure (i.e., steps, crystal planes, etc.), and charge transfer in the vicinity of the Pt sites (Nat. Catal. 2, 873–881 (2019)). Therefore, the prolonged introduction of CO alone significantly affects the chemical state of Pt on the surface of $\text{Pt}/\text{Sn}_{0.2}\text{Ti}_{0.8}\text{O}_2$ catalyst. Consequently, we were unable to perform in situ NAP XPS analysis on $\text{Pt}/\text{Sn}_{0.2}\text{Ti}_{0.8}\text{O}_2$ catalyst at different temperatures during the prolonged introduction of CO alone, as the active oxygen species of the catalyst continuously consumed.

However, simultaneous introduction of CO+O₂ ensures dynamic stability of the $\text{Pt}/\text{Sn}_{0.2}\text{Ti}_{0.8}\text{O}_2$ surface, thus we conducted in situ NAP-XPS analysis on $\text{Pt}/\text{Sn}_{0.2}\text{Ti}_{0.8}\text{O}_2$, $\text{Pt}/\text{TiO}_2\text{-R}$, and $\text{Pt}/\text{TiO}_2\text{-A}$ at different temperatures during the introduction of CO+O₂, as shown in Fig. 4d-f. The results indicate that only $\text{Pt}/\text{Sn}_{0.2}\text{Ti}_{0.8}\text{O}_2$ possessed the Pt^{4+} species after the introduction of CO+O₂, and the Pt^{4+} content increased with the increase of reaction temperature, suggesting that ROS was more significant with the increase of CO oxidation rate over $\text{Pt}/\text{Sn}_{0.2}\text{Ti}_{0.8}\text{O}_2$.

Fig. 2e CO₂ generation and corresponding CO concentration (inset) as a function of time during the transient CO oxidation without O₂ supply (1% CO/N₂) at 100 °C.

Figure 1 (for review only) *In situ* DRIFTS spectra recorded at 30–100 °C after Pt/Sn_{0.2}Ti_{0.8}O₂ exposed to 1% CO.

Fig. 4 | *In situ* DRIFTS and *in situ* NAP-XPS spectra of Pt/Sn_{0.2}Ti_{0.8}O₂, Pt/TiO₂-R and Pt/TiO₂-A recorded at 30–100 °C after exposure to CO+O₂. a–c, *In situ* DRIFTS spectra recorded at 30–100 °C after (a) Pt/Sn_{0.2}Ti_{0.8}O₂, (b) Pt/TiO₂-R and (c) Pt/TiO₂-A was exposed to 1% CO + 1% O₂ at a designated temperature under ambient pressure. d–f, *In situ* NAP-XPS Pt 4f spectra recorded at 30–100 °C after (d) Pt/Sn_{0.2}Ti_{0.8}O₂, (e) Pt/TiO₂-R and (f) Pt/TiO₂-A was exposed to 0.5 mbar of CO with 0.5 mbar of O₂ at a designated temperature.

4. It is hard to follow the way of XPS deconvolution. It was mentioned that the peaks at ~72.2 eV and ~74.8 eV could be assigned to 4f_{7/2} and 4f_{5/2} signals of Pt²⁺ species, whereas the peaks at ~74.4 eV and ~77.8 eV to 4f_{7/2} and 4f_{5/2} signals of Pt⁴⁺ species, respectively. These two peaks of 4f_{7/2} and 4f_{5/2} signals should have a similar number of FWHM (full width of half maximum), but they are randomly changing (see Fig. S3, etc.). This point needs to be clarified.

Response: Thank you very much for your suggestion. Following your advice, we have re-deconvoluted all the XPS spectra. During this process, we strictly fixed the peak area ratio of Pt 4f_{5/2} to 4f_{7/2} at 3:4, and set the FWHM of all Pt²⁺ and Pt⁴⁺ peaks to be identical. The resulting XPS spectra, after the re-deconvolution process, are shown in Figures 3a-c, 4d-f, S2b, and S3. Comparing with previous XPS deconvolution, the semiquantitative ratios of Pt⁴⁺ slightly decreases without changing the previous conclusion.

Fig. 3a–c *In situ* NAP-XPS Pt 4f spectra recorded at 100 °C after (a) Pt/Sn_{0.2}Ti_{0.8}O₂, (b) Pt/TiO₂-R and (c) Pt/TiO₂-A was exposed to 1 mbar of O₂, 0.5 mbar of CO with 0.5 mbar of O₂, and 1 mbar of CO in turn.

Fig. 4d–f *In situ* NAP-XPS Pt 4f spectra recorded at 30–100 °C after (d) Pt/Sn_{0.2}Ti_{0.8}O₂, (e) Pt/TiO₂-R and (f) Pt/TiO₂-A was exposed to 0.5 mbar of CO with 0.5 mbar of O₂ at a designated temperature.

Supplementary Fig. S2b Pt 4f XPS spectra of Pt/Sn_{0.2}Ti_{0.8}O₂ before and after pretreatment with H₂ at different temperatures.

Supplementary Fig. S3 Pt 4f XPS spectra over original and 300 °C H₂-pretreated (a) Pt/Sn_{0.2}Ti_{0.8}O₂, (b) Pt/TiO₂-R and (c) Pt/TiO₂-A.

Reviewer #2 (Remarks to the Author):

This manuscript by Li and coworkers address the mechanism of CO-induced oxygen spillover from the Sn-doped TiO₂ to supported Pt NPs using several in-situ techniques and DFT-based theoretical analyses. I think this is a comprehensive work and publishable. However, this manuscript does not satisfy the quality criteria for publication in Nature Communications. The bottom line is that the oxygen chemistry at the Pt-TiO₂ interfaces varies as Sn dopants are introduced. However, some vague experimental points make the authors' statements controversial.

Response: Thank you very much for your valuable comments, which allow us to further improve the manuscript. Hereinafter the one-by-one response to each specific comment as well as the revision accordingly.

Spillover of oxygen species from the supporting oxides to the supported metal NPs has always been considered in studying the catalytic functionality of the metal-oxide interfaces. Even in the cases of several early studies of CO oxidation catalyzed by oxide-supported metal NPs, reverse oxygen spillover has been considered a possible oxygen-supplying pathway. Moreover, the mechanism reported here by the authors is conceptually equivalent to oxygen taking over, which usually happens during the Mars-van Krevelen-type CO oxidation processes. I understand that the conceptual difference between the reverse oxygen spillover claimed by the authors with a conventional MvK type mechanism is whether the spillover oxygen can lay stable on the surface of Pt clusters or nanoparticles.

Response: Thank you for your careful review of our research. The CO catalytic oxidation process over Pt/TiO₂ catalyst generally follows the traditional MvK mechanism, where CO is adsorbed on Pt sites and then oxidized to CO₂ by lattice oxygen in the support. Similarly, CO oxidation on Pt/Sn_{0.2}Ti_{0.8}O₂ catalyst follows a MvK-like reaction pathway. The primary difference is that after CO adsorbs on the Pt site, it does not directly react with the lattice oxygen in the support. Instead, the lattice oxygen in the support migrates to the Pt site, where it reacts with the adsorbed CO to form CO₂. This migration process is known as reverse O spillover (ROS). In recent years, there has been a growing interest among researchers in studying the migration of oxygen on the catalyst surface. For instance, some related works on ROS are reviewed in Ref. ACS Catal. 2021, 11, 3159. Moreover, recent works by Hensen et al. (Refs. Nat. Catal. 2021, 4, 469 and Nat. Catal. 2020, 3, 526) and Vayssilov et al. (Ref.

Nat. Mater. 2011, 10, 310) have also mentioned the promoting effect of ROS on reactions over ceria related catalysts.

As for our manuscript, the principal innovation is that we modulated the rutile TiO₂ by Sn doping to activate low-temperature (< 100 °C) ROS in Pt/TiO₂ catalyst, and illustrated the rich interfacial chemistry of ROS from Sn-doped TiO₂ (SnTiO₂) to Pt sites in low-temperature CO oxidation with a combination of near-ambient-pressure XPS, *in situ* Raman/Infrared spectroscopies, and ab initio molecular dynamics (AIMD) simulations. We observed for the first time, to the best of our knowledge, the transformation of low-valent Pt²⁺ to high-valent Pt⁴⁺ with the presence of reducing gas CO, which indicates the CO-adsorption induced ROS on the catalyst.

I am skeptical whether the presence of Pt⁴⁺ directly confirms that oxygen was moved from the Sn-doped TiO₂ to Pt clusters. It is well-known that Pt clusters supported on a stoichiometric TiO₂ donate electrons to the support and that the Pt clusters draw electrons from the reduced TiO₂. Moreover, although the oxygen vacancy appeared to be associated with oxygen spillover, bulk oxygen can heal this vacancy thoroughly. If the local coordinative ensemble at the Pt-TiO₂ interfaces varies dynamically, I believe that the concentration of the Pt⁴⁺-like species may accordingly vary.

Response: We sincerely appreciate this remark from the reviewer, and find it especially helpful in improving the discussion of our manuscript. We would like to address our response as follows.

First of all, let's consider the well-known hydrogen spillover, the fundamentals of which have been well documented and widely applied in catalyst design. The hydrogen spillover process was depicted in Figure 1 in Ref. *ACS Catal.* 2021, 11, 3159: hydrogen molecules first adsorb and dissociate into hydrogen atoms on a metal surface; the hydrogen atoms then move across the metal surface to the metal-support interface, and they can diffuse across the support surface where hydrogen molecules cannot be dissociated. For the hydrogen spillover mechanism on a reducible M_xO_y support, Prins (Ref. Hydrogen spillover. Facts and fiction. *Chem. Rev.* 2012, 112, 2714) concluded that hydrogen molecules first adsorb and dissociate into hydrogen atoms on the metal surface, and hydrogen atoms then move to the metal-support interface where they become a combination of protons and electrons. The electrons reduce Mⁿ⁺ cations to M⁽ⁿ⁻¹⁾⁺ cations, and the protons bind to the surface oxygen anions. Therefore, the reduction of metal cations is an important indicator of hydrogen

spillover.

Figure 1 in Ref. ACS Catal. 2021, 11, 3159 Schematic illustration of hydrogen spillover and the associated processes

Then, compared with the hydrogen spillover process that leads to the reduction of the support, the reverse O spillover (ROS) process, in which the lattice oxygen transfers from the support to the nearby active site, results in accordingly the oxidation of the active site. Besides the reverse spillover of oxygen, other possibilities that may increase the valence of Pt species was discussed. In the section of Discussion in revised manuscript, we discussed three possibilities of Pt⁴⁺ generation only when Pt/Sn_{0.2}Ti_{0.8}O₂ was exposed to CO or CO+O₂. As shown below: "The *in situ* NAP-XPS spectra suggest that Pt²⁺ was the major Pt species of Pt/Sn_{0.2}Ti_{0.8}O₂, Pt/TiO₂-R and Pt/TiO₂-A after the moderate reduction, but only on Pt/Sn_{0.2}Ti_{0.8}O₂ catalyst that the generation of Pt⁴⁺ species was observed when it was exposed to CO or CO+O₂. Three possibilities may lead to the formation of Pt⁴⁺ on Pt/Sn_{0.2}Ti_{0.8}O₂: (i) Pt²⁺ captures gaseous O₂ thus being oxidized into Pt⁴⁺; (ii) Pt²⁺ donates electron to the support, therefore resulting in Pt⁴⁺ formation; (iii) lattice O in the support migrates to the vicinity of Pt²⁺ (i.e. the ROS process), causing the oxidation of Pt²⁺ to Pt⁴⁺. *In situ* NAP-XPS results (Fig. 3a) indicate that Pt⁴⁺ was not observed when Pt/Sn_{0.2}Ti_{0.8}O₂ was exposed to O₂ alone, eliminating the first possibility. Regarding the second possibility, it is well-known that Pt clusters supported on stoichiometric TiO₂ donate electrons to the support, and the Pt clusters draw electrons from the reduced TiO₂. Therefore, when reducing gas CO is introduced, Pt should receive electrons from the support, resulting in lower Pt valence. However, NAP-XPS (Fig. 3a) shows that when reducing gas CO was introduced, Pt²⁺ on the surface of Pt/Sn_{0.2}Ti_{0.8}O₂ is oxidized to Pt⁴⁺, indicating that electron transfer between Pt and the support cannot result in the Pt⁴⁺ formation. Therefore, the ROS process induced by CO adsorption should be the

cause of the oxidation of Pt^{2+} to Pt^{4+} ." Hence, we believe that the $\text{Pt}/\text{Sn}_{0.2}\text{Ti}_{0.8}\text{O}_2$ catalyst undergoes the process of O migration from the support to the active site (i.e. ROS) during the CO catalytic oxidation reaction.

Fig. 3a–c, *In situ* NAP-XPS Pt 4f spectra recorded at 100 °C after (a) $\text{Pt}/\text{Sn}_{0.2}\text{Ti}_{0.8}\text{O}_2$, (b) $\text{Pt}/\text{TiO}_2\text{-R}$ and (c) $\text{Pt}/\text{TiO}_2\text{-A}$ was exposed to 1 mbar of O₂, 0.5 mbar of CO with 0.5 mbar of O₂, and 1 mbar of CO in turn.

We also have indirect evidence to support the occurrence of ROS in the CO catalytic oxidation process of the $\text{Pt}/\text{Sn}_{0.2}\text{Ti}_{0.8}\text{O}_2$ catalyst. Yeung et al. (Ref. J. Am. Chem. Soc. 2021, 143, 196) proposed that Pt-based catalysts primarily follow the MvK mechanism during the high-temperature oxidation of toluene. Meanwhile, the large size of the toluene molecule necessitates a significant supply of oxygen during the oxidation process, rendering the ROS process's contribution to the process negligible. We evaluated the catalytic performance of $\text{Pt}/\text{Sn}_{0.2}\text{Ti}_{0.8}\text{O}_2$ and Pt/TiO_2 catalysts in the toluene catalytic oxidation reaction, and the findings revealed that the $\text{Pt}/\text{Sn}_{0.2}\text{Ti}_{0.8}\text{O}_2$ catalyst exhibited the lowest catalytic activity for toluene oxidation (see Figure 2 (for review only)), which indicates that the $\text{Pt}/\text{Sn}_{0.2}\text{Ti}_{0.8}\text{O}_2$ catalyst has the lowest reactivity of lattice oxygen. Therefore, the results indirectly indicate that the reason for the $\text{Pt}/\text{Sn}_{0.2}\text{Ti}_{0.8}\text{O}_2$ catalyst's superior performance in the CO catalytic oxidation process is due to the occurrence of the ROS process.

Figure 2 (for review only) Toluene oxidation performances of $\text{Pt}/\text{Sn}_{0.2}\text{Ti}_{0.8}\text{O}_2$, $\text{Pt}/\text{TiO}_2\text{-R}$ and

Pt/TiO₂-A.

We agree that the concentration of Pt⁴⁺-like species could fluctuate correspondingly with changes in the local coordination ensemble at the Pt-TiO₂ interfaces. For instance, we conducted in situ NAP XPS analysis on Pt/Sn_{0.2}Ti_{0.8}O₂, Pt/TiO₂-R, and Pt/TiO₂-A at different temperatures during the introduction of CO+O₂, as shown in Fig. 4d-f. The results indicate that only Pt/Sn_{0.2}Ti_{0.8}O₂ possessed the Pt⁴⁺ species after the introduction of CO+O₂, and the Pt⁴⁺ content increased with the increase of reaction temperature, suggesting that ROS was more significant with the increase of CO oxidation rate over Pt/Sn_{0.2}Ti_{0.8}O₂.

Fig. 4d–f, *In situ* NAP-XPS Pt 4f spectra recorded at 30–100 °C after (d) Pt/Sn_{0.2}Ti_{0.8}O₂, (e) Pt/TiO₂-R and (f) Pt/TiO₂-A was exposed to 0.5 mbar of CO with 0.5 mbar of O₂ at a designated temperature.

I am also not sure whether the surface of sub-nanometer-or nanometer (<2nm)-sized Pt clusters (experimentally observed ones) competitively binds oxygen rather than CO. This is presumably the key to whether the authors' claim can be rationalized or not. I am sure that the small Pt clusters strongly bind CO more than oxygen. In the cases of the larger Pt nanoparticles (than 2nm) in which micro-facets can develop at their surfaces, oxygen may share some portion of the surface with CO. However, it is hard to agree with the authors' claim that their small Pt clusters grip and stabilize oxygen at their surface. I agree that such oxygen can limitedly appear in the early stage of the reaction to which the catalyst is exposed to oxygen without CO. However, such oxygen species can be easily removed by CO (as presented in Fig. 5), and CO will preferentially occupy the surface of the Pt clusters. I am unsure whether an oxygen molecule heals the open position at stage v/VI (Fig. 5).

Response: After carefully consideration of your comment, we agree with the notion that small Pt clusters exhibit a stronger affinity towards CO compared to oxygen. Our DFT simulations also confirm this observation, as demonstrated in Supplementary

Figure S23, where the adsorption energy of O₂ on Pt/SnTiO₂ and Pt/TiO₂ catalyst surfaces is close to zero, indicating that O₂ is unlikely to adsorb onto small Pt clusters. Conversely, CO is more likely to adsorb onto small Pt clusters (as evidenced by the energy changes in steps I→II and i→ii of Figure 5e).

Supplementary Fig. S1 Adsorption energy of O₂ on Pt/Sn_{0.2}Ti_{0.8}O₂ and Pt/TiO₂-R.

Fig. 5e, Energy profiles and configurations of CO oxidation reaction following the pathway involving ROS (I-VIII) and that involving O_{latt} (i-vii).

We concur with the notion that it is difficult to capture and stabilize oxygen on the surface of small Pt clusters. Fig. 3a-c shows that after H₂ treatment, Pt/Sn_{0.2}Ti_{0.8}O₂, Pt/TiO₂-R, and Pt/TiO₂-A all mainly contained Pt²⁺ species on their surface. These Pt²⁺ species remained stable under O₂ atmosphere, indicating that small Pt clusters have limited capacity to trap and stabilize oxygen on their surface to generate higher valence states of Pt species. However, in contrast to the other catalysts, Pt²⁺ species in Pt/Sn_{0.2}Ti_{0.8}O₂ could be oxidized to Pt⁴⁺ in the presence of reducing gas CO (Fig. 3a), even without O₂ supply. This oxidation process is facilitated by the migration of

lattice O in the support to the Pt clusters (i.e. the ROS process). It is noteworthy that the ROS process is triggered by CO, which is subsequently oxidized by the O originating from the ROS process to produce CO₂ (the generation of CO₂ was evidenced by Fig. 2e). Thus, during CO oxidation, the small Pt clusters do not need to capture and stabilize gaseous oxygen on their surface. Gaseous O₂ only need to heal the oxygen vacancy on the support, which is generated by ROS process.

Fig. 3a–c *In situ* NAP-XPS Pt 4f spectra recorded at 100 °C after (a) Pt/Sn_{0.2}Ti_{0.8}O₂, (b) Pt/TiO₂-R and (c) Pt/TiO₂-A was exposed to 1 mbar of O₂, 0.5 mbar of CO with 0.5 mbar of O₂, and 1 mbar of CO in turn.

Fig. 2e CO₂ generation and corresponding CO concentration (inset) as a function of time during the transient CO oxidation without O₂ supply (1% CO/N₂) at 100 °C.

As for the comment "I am unsure whether an oxygen molecule heals the open position at stage v/VI (Fig. 5)." Generally, O₂ molecules tend to occupy oxygen vacancies on catalyst surfaces. In this study, Figure 5e shows that CO was oxidized by a lattice O on the support of Pt/Sn_{0.2}Ti_{0.8}O₂ and Pt/TiO₂-R catalysts, resulting in the release of a CO₂ molecule (steps ii→iv and I→V) and the creation of an oxygen vacancy on the support. O₂ is energetically more favourable to occupy this oxygen vacancy (as demonstrated by the energy changes in steps iv→v and V→VI of Figure 5e). The process of O₂ occupying oxygen vacancies during CO catalytic oxidation has been widely reported. For example, the description cited in Ref. Nat. Commun. 2020, 11, 1062: "CO reacts with the O on top of Pt ..., leaving one free coordination on Pt. This free coordination is then filled by adsorption of an O₂ molecule". And the

description cited in Ref. Nat. Chem. 2011, 3, 634: "The stoichiometric haematite surfaces near the Pt atoms are reduced partially to form an oxygen vacancy (step i), which can adsorb the O₂ reactants." Therefore, we believe that the process of "an oxygen molecule healing the open position at stage v/VI (Fig. 5e)" is reasonable.

Moreover, I think assigning the PtO and Pt-O-Ti separately in XAS, or XPS spectra is risky because the local atomic ensemble at the metal-oxide is complicated, and dynamic electronic variation occurs during the reaction (as reported by Daelman et al. Nat. Mater. 2019,18, 1215).

Response: In XPS spectra, we only assigned Pt species to Pt²⁺ and Pt⁴⁺ (Fig. 3a–c and Fig. 4d-f). In XAS spectra, the fitting results were listed in Table S4. No Pt-O-Ti was assigned in the fitting results. We are unsure if you were referring to Pt-O-Pt and Pt-O-Sn. However, it should be noted that "Pt-O-Sn" or "Pt-O-Pt" are terms used to describe a part of the micro-structure surrounding Pt, in which either Sn or Pt atom is bound to the central Pt atom through O. The bond distances of Pt-O-Sn (3.00 Å) and Pt-O-Pt (3.56 Å) are significantly longer than that of Pt-O (~2.00 Å). Therefore, we believe that it is reasonable to assign Pt-O, Pt-O-Sn and Pt-O-Pt separately in XAS.

Supplementary Table S1 EXAFS fitting results of Pt L₃-edge over Pt/Sn_{0.2}Ti_{0.8}O₂, Pt/TiO₂-R and Pt/TiO₂-A. CN: coordination numbers; R: bond distance; σ²: Debye-Waller factors; ΔE₀: the inner potential correction. R factor: goodness of fit. S₀² was set to 0.86, according to the experimental EXAFS fit of Pt foil reference by fixing CN as the known crystallographic value.

Sample	shell	CN	R(Å)	σ ²	ΔE ₀	R factor
	Pt-O	4.5±0.3	1.99±0.03	0.0092		
Pt/Sn _{0.2} Ti _{0.8} O ₂	Pt-Pt	1.4±0.5	2.70±0.03	0.0021	11.7±1.7	0.0159
	Pt-O-Sn	1.1±0.4	3.00±0.03	0.0019		
Pt/TiO ₂ -R	Pt-O	3.4±0.3	2.03±0.01	0.0017	17.8±1.6	0.0179
	Pt-Pt	1.4±0.5	2.79±0.02	0.0027		
	Pt-O	4.1±0.3	2.01±0.02	0.0065		
Pt/TiO ₂ -A	Pt-Pt	1.2±0.4	2.73±0.03	0.0019	14.8±1.8	0.0103
	Pt-O-Pt	3.4±1.4	3.56±0.05	0.0022		

Fig. 5 shows the overall CO oxidation pathway following the Langmuir-Hinshelwood mechanism. This is, unfortunately, equivalent to the case of CO oxidation by Pt-oxide clusters, as an oxygen molecule is required to restore the initial Pt-oxide cluster.

Response: As described in Ref. Nat. Commun. 2015, 6, 8675: "The generally accepted by platinum is one of Langmuir-Hinshelwood (LH) mechanism requires the adsorption and reaction of molecular CO with atomic oxygen over metallic platinum surfaces". It suggests that Langmuir-Hinshelwood mechanism of CO catalytic oxidation is the reaction between adsorbed CO and adsorbed O₂ molecule. As you mentioned, Fig. 5 shows that CO was oxidized by Pt-oxide clusters. This means CO was oxidized by lattice O, i.e. following the MvK mechanism (Ref. Nat. Commun. 2019, 10, 1358, which said that "It has also been observed that the relevant step during CO oxidation in a MvK reaction mechanism is the reaction between CO adsorbed on Pt and oxygen from the lattice"). Therefore, we believe Fig. 5 mainly shows the overall CO oxidation pathway following the MvK mechanism.

Fig. 5e, Energy profiles and configurations of CO oxidation reaction following the pathway involving ROS (I-VIII) and that involving O_{latt} (i-vii).

On page 16, "The energy barrier of CO oxidation by ROS over Pt/Sn_{0.2}Ti_{0.8}O₂ was 0.69 eV, which was obviously lower than that of CO oxidation by O_{latt} (0.90 eV), suggesting that CO oxidation by the O atom transferred through ROS is more preferred over Pt/Sn_{0.2}Ti_{0.8}O₂." I think this kind of statement made from the DFT calculations on flat TiO₂ surfaces cannot be generalized. The barrier for the direct

MvK mechanism will decrease and become sensitive to the surface morphology of TiO₂, which is the case of the experimentally synthesized catalyst.

Response: Thank you very much for your suggestions. Fig. 5e depicts a complete CO oxidation reaction pathway: First, CO was oxidized by lattice oxygen, leaving an oxygen vacancy; Second, the oxygen vacancy was filled by an O₂ molecule, forming an O-O bond on the surface; Third, another CO was oxidized by the O-O bond and consumed one O atom, and then the remaining O atom healed the oxygen vacancy, restoring the catalyst to its initial state. Therefore, in this reaction cycle, we have considered the reaction pathway and energy changes of CO catalytic reaction on the catalyst surface with vacancies. Certainly, we understand that the catalyst model used in DFT simulations often differs from the actual situation, and DFT is also difficult to simulate all actual reaction processes. Therefore, we have revised this description to indicate that only in this DFT simulation, the result of a lower energy barrier for the ROS reaction pathway was obtained. The revised description is as follows:

"The results of this DFT simulation show that the energy barrier of CO oxidation by ROS over Pt/Sn_{0.2}Ti_{0.8}O₂ was 0.69 eV, which was lower than that of CO oxidation by O_{latt} (0.90 eV), suggesting that CO oxidation by the O atom transferred through ROS is probably more preferred over Pt/Sn_{0.2}Ti_{0.8}O₂."

Reviewer #3 (Remarks to the Author):

In this work, the authors synthesized Pt-based catalysts supported on $\text{Sn}_x\text{Ti}_{1-x}\text{O}_2$, TiO_2 (anatase) and TiO_2 (rutile) supports. Catalyst structures were thoroughly characterized by electron microscopy, XRD, AP-XPS, DRIFTS and in-situ RAMAN. The catalytic activity was probed by CO oxidation, and it was found that $\text{Pt}/\text{Sn}_x\text{Ti}_{1-x}\text{O}_2$ exhibited improved low temperature CO oxidation than Pt/TiO_2 (anatase and Pt/TiO_2 (rutile). Therefore, the authors devoted extended efforts to understand the nature of such improved low temperature activity promotion when Sn was used to dope TiO_2 . Overall, Pt particle size was similar across all samples and CO-TPR experiments revealed improved lattice oxygen availability on $\text{Pt}/\text{Sn}_x\text{Ti}_{1-x}\text{O}_2$. AP-XPS and in-situ RAMAN were critical to identify that upon the exposure of $\text{Pt}/\text{Sn}_x\text{Ti}_{1-x}\text{O}_2$ to $\text{CO}+\text{O}_2$, increased Pt^{4+} species and weaker Ti-O-Sn bonds were detected, which suggests lattice oxygen from Ti-O-Sn could be mobilized towards Pt centers (process called Reverse Oxygen Spillover by the authors (ROS)). Computational chemistry calculations (AIMD) illustrated that ROS occurs on model $\text{Pt}_4\text{O}_4/\text{Ti-O-Sn}$ and that lowest energy barrier CO oxidation pathway occurs via ROS rather than other routes. I recommend this article for publications only if the authors can satisfactorily address the following concerns, clarifications, and improvements to the manuscript.

Response: Thank you very much for your positive overall evaluation of our work. We responded to your specific comments individually in the following sections.

1. Novelty

Q1.1. Reverse spill over of oxygen has recently been reported in JACS (J. Am. Chem. Soc. 2023, 145, 2523–2531). Such publication is crucially relevant for the present manuscript and should be included in the introduction. It also raises the question on the novelty of this manuscript. Authors should clearly clarify the novelty of this manuscript or what additional knowledge or novelty this manuscript brings over what already been reported in the JACS report.

Response: Thank you for providing us with an important recently-published reference. After careful examination, we found that the core content of this article is the synthesis of single-atom catalysts (SACs) Pt/CeO_2 through treatment under O_2 and N_2 conditions. However, the CO catalytic oxidation activity of SAC Pt/CeO_2 catalysts obtained through these two treatment methods differs significantly. Through Raman

spectroscopy and computational studies, the authors revealed the distribution of various $\text{Pt}_1\text{O}_n\text{-Ce}^{\delta+}$ species in each specific SACs, and found that the minority species of $\text{Pt}_1\text{O}_4\text{-Ce}^{3+}\text{-O}_v$, accounting for only 14.2%, affords the highest site-specific reactivity for low-temperature CO oxidation among the other abundant counterparts, i.e., $\text{Pt}_1\text{O}_4\text{-Ce}^{4+}$ and $\text{Pt}_1\text{O}_6\text{-Ce}^{4+}$. This work elucidates the quantitative distribution and dynamic transformation of varied single-atom species in a given SAC, offering a more intrinsic descriptor and quantitative measure to depict the inhomogeneity of SACs.

In this literature, the authors speculate that the formation of $\text{Pt}_1\text{O}_4\text{-Ce}^{3+}\text{-O}_v$ derived from the reverse O spillover (ROS) during the synthesis process, but there is little visual experimental evidence. The original text is described as follows: "However, based on our results, the lattice oxygen also promotes the oxidation and dispersion of Pt atoms, likely via reverse oxygen spillover.^{12,17,24} This is further confirmed by the higher $\text{Ce}^{3+}/\text{Ce}_{\text{total}}$ value (20.2%) in Pt/CeO₂-N600 than those of Pt/CeO₂-O600 (15.3%) and Pt/CeO₂-O800 (13.4%) (Figure 2c and Table S1). The absence of reduction peaks at low temperatures (269 and 275 °C for Pt/CeO₂-N600 and Pt/CeO₂-N800, respectively) indicates that the lattice oxygen at the interface are readily consumed during the nonoxidative dispersion by reverse spillover from Ce^{4+} to Pt SACs²⁵⁻²⁷ (Figure 2d)."

Figure 2 in Ref. J. Am. Chem. Soc. 2023, 145, 2523–2531 (c) Contents of different Pt and Ce species based on the XPS analysis, and the coordination number of Pt-O obtained from EXAFS analysis. (d) H₂-TPR results of the Pt/CeO₂ catalysts treated by different temperatures and atmospheres.

As for our manuscript, we modulated the rutile TiO₂ by Sn doping to activate

low-temperature ($< 100\text{ }^{\circ}\text{C}$) ROS in Pt/TiO₂ catalyst, and illustrated the rich interfacial chemistry of ROS from Sn-doped TiO₂ (SnTiO₂) to Pt sites in low-temperature CO oxidation with a combination of near-ambient-pressure XPS, *in situ* Raman/Infrared spectroscopies, and ab initio molecular dynamics (AIMD) simulations. We observed for the first time, to the best of our knowledge, the transformation of low-valent Pt²⁺ to high-valent Pt⁴⁺ with the presence of reducing gas CO. In conclusion, the innovation of our manuscript is different from that of the JACS report.

Based on your suggestion, we have included this article in our Introduction section. As shown below:

"The concept of strong metal support interaction (SMSI) effect has been widely used to describe and/or interpret phenomena of electronic interaction, as well as the stabilization/destabilization of metals on support materials⁵⁻⁷. For instance, Ding et al.⁸ synthesized single-atom Pt/CeO₂ catalysts via oxidative and non-oxidative dispersions, and found significant differences in the CO catalytic oxidation activity of single-atom Pt in different coordination environments."

Q1.2. The authors should provide clear rationales on why Sn was selected to dope TiO₂ instead of any other promoters. What are the reasons/hypothesis/preliminary work behind such selection? What is the generality of selecting a promoter such as Sn to achieve such performances.

Response: Thank you for bringing the logical flaw in our Introduction to our attention. We selected Sn as dopant because SnO₂ has similar crystal structure as rutile TiO₂. Therefore, doping Sn will not significantly change the bulk structure but alter the oxygen symmetry in TiO₂. It is well-documented that the creation of asymmetric oxygen will increase its mobility and thus benefits to the ROS process. In the revised manuscript, we describe the rationale for selecting Sn-doped TiO₂ as the support in the Introduction as follows:

"Since SnO₂ possess a similar structure as rutile TiO₂ (ref), doping Sn will not change significantly the bulk construction but create asymmetric oxygens (M₁-O-M₂) in TiO₂. Such altering of oxygen symmetry probably increases its mobility and thus benefits to the ROS process."

2. Clarifications

Q2.1. Line 86. The author should use the term USDRIVE, not USDRIVER. (Please

see the following reference: <https://www.energy.gov/eere/vehicles/us-drive>)

Response: Thank you for your suggestion. The term "USDRIVER" has been corrected to "USDRIVE" in the revised manuscript.

Q2.2. How were TOFs estimated? I could not find how the fraction of active Pt species was estimated for the TOF calculations. Please clarify and provide a detailed explanation of methods.

Response: We appreciate your comment. Since Pt is a precious metal with considerable price, we took all the Pt atoms into consideration when calculating the TOF_{Pt} in our work. We believe such calculation is reasonable so that the activity of the catalyst can be compared under the same Pt usage (i.e., similar cost). Of course, this calculation method will underestimate the TOF_{Pt} of the catalyst. Even though, the Pt/ $\text{Sn}_{0.2}\text{Ti}_{0.8}\text{O}_2$ catalyst reported in this study still showed a higher TOF_{Pt} compared to Pt-based catalysts reported in the literatures.

Following your comments, we have added a specific description of calculating the Pt content in the revised manuscript, as shown below:

$$\text{TOF}_{\text{Pt}} = \frac{X_{\text{CO}} \cdot F_{\text{CO}}}{N_{\text{Pt}}} \times 100\% \quad (3)$$

where, X_{CO} is CO conversion (< 20%), F_{CO} ($\mu\text{mol s}^{-1}$) is molar gas flow rate of CO, and N_{Pt} (μmol) is the total number of Pt atoms on the catalyst. It should be noted that all the catalysts used in this study contained 0.5 wt% Pt. Moreover, all Pt atoms were considered when calculating TOF_{Pt} to enable a fair comparison of catalyst activity under the same Pt usage."

Q3.3. Include detailed CO-TPR (shown in Figure 2e) experiment description in methods section. To assess the validity of author's claim about correlation between active species in the CO-TPR and O₂-TPD experiments (shown in figure 2e and 2f) it is crucial for reviewers to understand exactly how the CO-TPR was carried out, including any pre-treatments with O₂ or purge of weakly adsorbed O₂ species. Since the main finding of this paper (Reverse Oxygen Spillover on Ti/Sn_xTi_{1-x}O₂) is based CO-TPR and O₂-TPD experiment analysis (lines 222-226), I highly suggest that further clarification of experiments is provided before this work can be considered for publication.

Response: We apologize for the confusion caused by the description of the experimental conditions of Figure 2e. Figure 2e depicts a transient reaction with a fixed reaction temperature of 100 °C and CO flowing without O₂ (1% CO/N₂). The purpose of this experiment is to investigate how many active oxygen species on the catalyst surface can participate in the reaction under the temperature conditions of CO oxidation. Therefore, the reduced catalysts were loaded into the reactor and heated to the reaction temperature under N₂ without any further pretreatment. The same experiment was also conducted at 200 °C (see Supplementary Fig. S19), and the results were consistent with those at 100 °C. Both experiments demonstrated that the Pt/Sn_{0.2}Ti_{0.8}O₂ catalyst has the most active oxygen species during CO oxidation, which cannot be characterized by O₂-TPD (as shown in Figure 2f). Following your suggestion, we have added a detailed description of the experimental conditions of this transient reaction in the Methods section, hoping to clarify your question. As shown below:

"Transient CO oxidation was tested in a fixed-bed quartz micro-reactor. The reduced catalysts were loaded into the reactor and heated to the reaction temperature under N₂ without any further pretreatment. The reaction temperature was fixed at 100 or 200 °C with a CO concentration of 1% and N₂ as the balance gas, without the supply of O₂. The gas flow rate was set to 100 mL·min⁻¹ with a GHSV of 60,000 mL g_{cat}⁻¹ h⁻¹. The infrared gas analyzer (Gasetm Dx-4000) was utilized to measure the concentrations of CO and CO₂ in both the inlet and outlet streams."

Fig. 2 e, CO₂ generation and corresponding CO concentration (inset) as a function of time during the transient CO oxidation without O₂ supply (1% CO/N₂) at 100 °C. **f**, O₂-TPD profiles.

Supplementary Fig. S2 CO₂ generation and corresponding CO concentration (inset) as a function of time during the transient CO oxidation without O₂ supply (1% CO/N₂) at 200 °C.

Q3.4. It would be highly appreciated if authors could indicate which Sn-O-Ti bonds are broken in CO oxidation cycle on the ROS route to facilitate understanding of reaction cycle schemes.

Response: The Sn-O-Ti bonds broke in CO oxidation cycle on the ROS route could be observed by the AIMD simulation (see Fig. 5c,d and the Supplementary Video). As described in our manuscript: "Fig. 5c and d show the variation of atomic distance between Ti and O (Ti-O), Sn and O (Sn-O), and Pt and O (Pt-O) as a function of simulation time over Pt/Sn_{0.2}Ti_{0.8}O₂ and Pt/TiO₂-R. The atomic distance of Ti-O, Sn-O and Pt-O stays at 2~3 Å during the first 6 ps of AIMD simulation (Fig. 5c), indicating that the O in Sn-O-Ti kept bonding with the Pt of Pt₄O₄ clusters. Then, the atomic distance of Ti-O and Sn-O quickly increased at 6–9 ps, while the atomic distance of Pt-O remained unchanged. The result indicates cleavage of Ti-O and Sn-O bonds in Sn-O-Ti, subsequently leading to the reverse flow of O towards the Pt sites." Therefore, the Sn-O-Ti bonds that contribute to the ROS pathway specifically refer to the Sn-O-Ti connected to the Pt site through O, as shown in the upper left corner of Figure 5c.

We have also incorporated this description into the relevant section of Discussion, as follows: "Structure optimization suggests that the Pt₄O₄ clusters tended to connect with Sn-O-Ti through the O site in Sn-O-Ti. The charge density of O in these Sn-O-Ti bonds increased after the loading of Pt₄O₄ clusters as well as the subsequent adsorption of CO. We speculated that the increased charge density promoted the mobility of the O in Sn-O-Ti, resulting in the occurrence of ROS in Pt/Sn_{0.2}Ti_{0.8}O₂ during CO oxidation reaction."

Fig. 5c–d, Atomic distance of Ti-O, Sn-O and Pt-O as a function of simulated time during AIMD simulation over (c) Pt/Sn_{0.2}Ti_{0.8}O₂ and (d) Pt/TiO₂-R.

3. Improvements

Q3.1. The discussion on why several reduction treatments were done to finally select 300 °C in H₂ is not scientifically interesting/relevant to the main goal of the study (understanding the ROS effect) and should be placed in the SI section.

Response: Thank you for your suggestion. The discussion regarding to reduction treatments was placed into the **Supplementary Note 2** in Supporting Information. As shown below:

"The H₂ pretreatment process utilized a 5% H₂ concentration and a treatment time of 1 h. Subsequently, Pt/Sn_{0.2}Ti_{0.8}O₂ was subjected to various treatment temperatures. Supplementary Fig. S2a demonstrates that Pt/Sn_{0.2}Ti_{0.8}O₂ treated at 300 °C exhibited the highest CO oxidation performance. Therefore, the designed Pt/Sn_xTi_{1-x}O₂ and Pt/TiO₂ catalysts were evaluated for CO oxidation after pretreatment in 5% H₂ at 300 °C for 1 h."

Q3.2. Indicate error bars in figures S5B and S5C (data set in Arrhenius plots) as well as in all figures containing TOF measurements.

Response: The error bars have been added into all figures containing TOF measurements and Arrhenius plots. As shown below:

Fig. 1b Turnover frequencies of CO oxidation over Pt sites (TOF_{Pt}) measured at 80 °C and 100 °C with CO conversion below 20%.

Supplementary Fig. S4b TOF_{Pt} measured at 80 °C and 100 °C with CO conversion below 20%.

Supplementary Fig. S3 (b) Arrhenius plots measured at 50–130 °C with CO conversion below 20%. (c) TOF_{Pt} measured at 80 °C and 100 °C with CO conversion below 20%.

Supplementary Fig. S6 Arrhenius plots measured at 50–130 °C with CO conversion below 20% over Pt/Sn_xTi_{1-x}O₂ and Pt/TiO₂ catalysts with H₂ pretreatment.

Q3.3. Lines 126. I recommend including a more elaborated explanation on why the observed CO and O₂ partial orders are ~zero. The following work can help with that *Angew.Chem.Int.Ed.*2021,60,26054–26062.

Response: Thank you for your comment. We have thoroughly reviewed the literature you mentioned. Based on the contents of the references, we have included an explanation for why the CO and O₂ partial orders are close to 0, and have cited the references you mentioned in revised manuscript. Specifically, the details are as follows:

"Meanwhile, the reaction order of CO and O₂ during CO catalytic oxidation over Pt/Sn_{0.2}Ti_{0.8}O₂, Pt/TiO₂-R and Pt/TiO₂-A were all slightly higher than 0 (Supplementary Fig. S8), indicating that CO oxidation over these three Pt-based catalysts follow the Mars–van Krevelen (MvK) mechanism, which is typical for reducible oxide-based catalysts. Additionally, the partial orders of CO (or O₂) were approximately 0, highlighting the weakened kinetic relevance of CO (or O₂) adsorption/activation over Pt/Sn_{0.2}Ti_{0.8}O₂, Pt/TiO₂-R and Pt/TiO₂-A."

Q3.4. Lines 137-145. Is the observed sulfur tolerance expected for Pt- and TiO₂-based materials? If SO₂-tolerance is not novel due to Sn doping (I don't think so), discussion of relevant literature is appropriate. (example: *NATURE CATALYSIS* | VOL 2 | JULY 2019 | 614–622)

Response: Pt-based materials generally exhibit poor sulfur resistance in CO catalytic oxidation reactions. Hence, the observed superior sulfur tolerance of Pt/Sn_{0.2}Ti_{0.8}O₂ is not expected. Sn doping may hold innovative potential for enhancing the sulfur resistance of Pt/TiO₂ catalysts.

However, since it is not the main focus of this paper, we did not delve into it extensively. Nonetheless, we are grateful for the valuable reference provided, which enabled us to briefly discuss how Sn doping enhances the sulfur resistance of Pt/TiO₂ catalysts in the main text. This discussion also laid the foundation for our subsequent exploration of the significant improvement in sulfur resistance achieved through Sn doping in Pt/TiO₂ catalysts:

"One plausible explanation for the enhanced sulfur resistance is that the

introduction of Sn doping results in a notable modification of the coordination environment of Ti within the support structure, thereby influences the interaction between the active Pt site and the support, leading to improved sulfur resistance⁴⁴."

REVIEWERS' COMMENTS

Reviewer #1 (Remarks to the Author):

After thoroughly reviewing the revised manuscript and the corresponding reply letter, I find the authors' response to my comments and criticism to be appropriate. Consequently, the quality and presentation of the paper have been significantly improved. In light of this, I recommend that the paper be accepted in its current form.

Reviewer #2 (Remarks to the Author):

The authors addressed most of my concerns.

Comments:

1. My point in discussing competitive adsorption of CO and oxygen on Pt cluster is whether the surface of such small Pt clusters could be initially saturated by oxygen, as presented in Fig. 5. This should be when the Pt cluster relatively strongly binds oxygen then CO. How about comparing the sequential and competitive binding energy of CO and O-atom on the surface of small Pt clusters and evaluate whether the initial structure i/l is energetically feasible? I think it is a possible assumption that the initial Pt structure for DFT calculations could be the one with saturated COs on top of Pt atoms and additional oxygen atoms at the Pt-Pt bridging sites or Pt hollow sites. Of course, this is the only case where the small Pt cluster binds CO and O well together.
2. The results are based on a specific Sn-doped TiO₂. The authors may discuss how the results can be generalized over reducible oxide supports. Alternatively, a design rule of better-performing catalysts according to the authors' findings should be discussed.

Response to referees

Reviewer #1 (Remarks to the Author):

After thoroughly reviewing the revised manuscript and the corresponding reply letter, I find the authors' response to my comments and criticism to be appropriate. Consequently, the quality and presentation of the paper have been significantly improved. In light of this, I recommend that the paper be accepted in its current form.

Response: We sincerely appreciate the valuable improvements you have made to the quality of our manuscript through your review comments. Thank you very much for your diligent efforts during the review process.

Reviewer #2 (Remarks to the Author):

The authors addressed most of my concerns.

Comments:

1. My point in discussing competitive adsorption of CO and oxygen on Pt cluster is whether the surface of such small Pt clusters could be initially saturated by oxygen, as presented in Fig. 5. This should be when the Pt cluster relatively strongly binds oxygen then CO. How about comparing the sequential and competitive binding energy of CO and O-atom on the surface of small Pt clusters and evaluate whether the initial structure i/I is energetically feasible? I think it is a possible assumption that the initial Pt structure for DFT calculations could be the one with saturated COs on top of Pt atoms and additional oxygen atoms at the Pt-Pt bridging sites or Pt hollow sites. Of course, this is the only case where the small Pt cluster binds CO and O well together.

Response: We sincerely appreciate your contribution in providing an alternative perspective that allows us to reexamine the process of DFT simulation. In this study, SAC-STEM images and XPS results revealed that the Pt species on Pt/Sn_{0.2}Ti_{0.8}O₂ and Pt/TiO₂-R existed in the form of small PtO clusters. Therefore, for the DFT simulation process, Pt₄O₄ clusters were placed on the surface of support Sn_{0.2}Ti_{0.8}O₂ and TiO₂-R. To validate the stability of the configurations of Pt/Sn_{0.2}Ti_{0.8}O₂ and Pt/TiO₂-R, these configurations underwent a 10 ps AIMD simulation at 700 K before the formal DFT simulation process. Then, the configurations at 2, 4, 6, 8, and 10 ps of

AIMD simulation were subsequently used as initial structures and optimized using VASP to obtain the most stable configurations of Pt/Sn_{0.2}Ti_{0.8}O₂ and Pt/TiO₂-R. The total energy of these optimized configurations is listed in Supplementary Table S5. Notably, the total energy of the configurations after different AIMD simulation periods was nearly identical, indicating that the initial structure i/I is energetically feasible.

It is important to note that the initial structure refers to the PtO clusters loaded on the support, rather than the structure formed after Pt is loaded on the support and then oxygen is adsorbed. Hence, the initial structure i/I only confirms the stability of PtO clusters on the catalyst surfaces, and does not imply a strong binding between the Pt cluster and oxygen compared to CO. These configurations we constructed is consistent with the results obtained from SAC-STEM and XPS characterization. After obtaining catalyst structures that closely resemble the actual state, we performed DFT simulations to investigate the O₂ adsorption capability of this structure. The results (see Supplementary Fig. S23) indicate that the surface of this structure exhibits negligible O₂ adsorption. Therefore, we believe that the reaction cycle presented in Fig. 5, starting with CO adsorption, is reasonable.

Supplementary Table S5 Total energy optimized by VASP using the initial configurations taken from 2–10 ps of AIMD simulation.

Sample	2 ps/eV	4 ps/eV	6 ps/eV	8 ps/eV	10 ps/eV
Pt/Sn _{0.2} Ti _{0.8} O ₂	-1917.73	-1917.72	-1917.73	-1917.73	-1917.73
Pt/TiO ₂ -R	-2022.77	-2022.73	-2022.79	-2022.80	-2022.73

Supplementary Fig. S23 Adsorption energy of O₂ on Pt/Sn_{0.2}Ti_{0.8}O₂ and Pt/TiO₂-R.

2. The results are based on a specific Sn-doped TiO₂. The authors may discuss how the results can be generalized over reducible oxide supports. Alternatively, a design rule of better-performing catalysts according to the authors' findings should be discussed.

Response: We sincerely appreciate your valuable suggestions. However, this study did not specifically reveal the underlying principles responsible for the pronounced ROS phenomenon in Sn-doped TiO₂. We speculate that this could be attributed to the strong oxygen migration ability of SnO_x itself, as well as the asymmetric oxygen structure formed by Sn doping. In conclusion, enhancing the oxygen migration ability of support is an effective means to promote ROS activity. Following your advice, we have added relevant discussions in the Discussion section as follows: "Overall, we activated the low temperature reverse oxygen spillover on Titania-supported Platinum catalyst by introducing Sn into TiO₂ support, and further demonstrated the existence and mechanistic route of reverse oxygen spillover in low-temperature CO oxidation by a combination of experimental and theoretical studies. The revealed interfacial dynamics in reverse oxygen spillover fills the gaps of interfacial chemistry of adsorbates and/or intermediate transport through metal-support interfaces, and allows deeper fundamental understanding of catalytic reactions involving oxygen, and hence improvement of catalyst design for technologically relevant redox reactions. We speculate the strategy to improve the oxygen mobility of support, such as the construction of asymmetric oxygens (M₁-O-M₂) or doping secondary metals with weak M-O bond, probably is particularly effective to promote ROS. Looking forward, we anticipate that the reactant-adsorption-triggered characteristics of oxygen spillover will arouse further investigations on the mechanistic effect of microstructures (such as size of active sites, nature of supports, degree of O charging) and reactants (such as hydrocarbons, H₂O and SO₂), finding more potential applications in thermo-, photo- and electro-catalysis."